# Age-dependent loss of adipose Rubicon promotes metabolic disorders via excess autophagy

Tadashi Yamamuro [1], Tsuyoshi Kawabata[1,2,3], Atsunori Fukuhara [4,5], Shotaro Saita[1,2], Shuhei Nakamura [1,2,6], Hikari Takeshita[7], Mari Fujiwara[1], Yusuke Enokidani[1], Gota Yoshida[1], Keisuke Tabata [1,2], Maho Hamasaki[1,2], Akiko Kuma[1,2], Koichi Yamamoto[7], Iichiro Shimomura[4✉] & Tamotsu Yoshimori[1,2,8✉]

The systemic decline in autophagic activity with age impairs homeostasis in several tissues, leading to age-related diseases. A mechanistic understanding of adipocyte dysfunction with age could help to prevent age-related metabolic disorders, but the role of autophagy in aged adipocytes remains unclear. Here we show that, in contrast to other tissues, aged adipocytes upregulate autophagy due to a decline in the levels of Rubicon, a negative regulator of autophagy. *Rubicon* knockout in adipocytes causes fat atrophy and hepatic lipid accumulation due to reductions in the expression of adipogenic genes, which can be recovered by activation of PPARγ. SRC-1 and TIF2, coactivators of PPARγ, are degraded by autophagy in a manner that depends on their binding to GABARAP family proteins, and are significantly downregulated in *Rubicon*-ablated or aged adipocytes. Hence, we propose that age-dependent decline in adipose Rubicon exacerbates metabolic disorders by promoting excess autophagic degradation of SRC-1 and TIF2.

[1] Department of Genetics, Graduate School of Medicine, Osaka University, Osaka, Japan. [2] Laboratory of Intracellular Membrane Dynamics, Graduate school of Frontier Biosciences, Osaka University, Osaka, Japan. [3] Department of Stem Cell Biology, Atomic Bomb Disease Institute, Nagasaki University, Nagasaki, Japan. [4] Department of Metabolic Medicine, Graduate School of Medicine, Osaka University, Osaka, Japan. [5] Department of Adipose Management, Graduate School of Medicine, Osaka University, Osaka, Japan. [6] Institute for Advanced Co-Creation Studies, Osaka University, Osaka, Japan. [7] Department of Geriatric and General Medicine, Graduate School of Medicine, Osaka University, Osaka, Japan. [8] Integrated Frontier Research for Medical Science Division, Institute for Open and Transdisciplinary Research Initiatives (OTRI), Osaka University, Osaka, Japan. ✉email: ichi@endmet.med.osaka-u.ac.jp; tamyoshi@fbs.osaka-u.ac.jp

Metabolic disorders, and the insulin resistance associated with them, represent a worldwide health concern due to their increasing incidence in developed countries and their contribution to life-threatening cardiovascular diseases[1,2]. Adipose tissue, especially white adipose tissue (WAT), regulates systemic metabolism via two functions[3,4]. Adipocytes store surplus energy as triglyceride in a single monolayer organelle, the lipid droplet (LD)[5,6], and they also orchestrate other metabolic organ functions through the secretion of several hormones or cytokines (called adipocytokines), such as leptin and adiponectin[7]. Because diabetes, fatty liver and dyslipidaemia are closely associated with aging, a mechanistic understanding of age-associated metabolic complications is of particular interest. The connection between adipose tissue and aging is illustrated by age-associated systemic changes in adipose tissue such as ectopic lipid deposition and adipose-derived hormone changes[8]. However, the direct cause of adipocyte dysfunction in aged organisms, which in turn leads to age-associated metabolic disorders, remains unknown.

Autophagy is an intracellular bulk degradation system that plays a pivotal role in the maintenance of cellular homoeostasis and the prevention of a wide range of age-associated diseases, such as neurodegeneration[9,10], hepatic disorders[11,12] and heart failure[13]. Consistent with this, recent advances in the field have shown that autophagy is crucial for both adipose tissue homoeostasis and longevity[14]. Autophagy, especially mitophagy, regulates adipocyte differentiation[15,16] or beige adipocyte maintenance[17,18], whereas a form of selective autophagy, lipophagy, degrades LDs in brown adipose tissue (BAT) and liver[19–21]. The age-associated decline in autophagic activity[22,23], along with the extension of longevity by upregulation of basal autophagy[24,25], indicates that autophagy is involved in the aging process. Rubicon negatively regulates autophagy by binding to the PI3K complex[26–28], which is essential for autophagy. Our recent studies have revealed that Rubicon levels increase with obesity or aging, and that accumulation of this factor suppresses autophagy and disrupts cellular homoeostasis in several tissues[29,30]. However, the role of Rubicon in adipocytes, especially in aging, remains largely unknown. In addition, it is important to determine whether, and if so how, the age-associated change in adipose autophagy causes metabolic diseases.

Here, we report that an age-dependent loss of Rubicon and an increase in autophagic activity are key feature of aged adipocytes. We found that genetic ablation of Rubicon in adipocytes causes inadequate upregulation of autophagy, which in turn disrupts proper adipocyte functions, leading to fat atrophy, glucose intolerance, dyslipidaemia and hepatic fat accumulation. These changes arise from a reduction in adipogenic gene expressions, which can be restored by activation of PPARγ. SRC-1 and TIF2, coactivators of PPARγ, are degraded by autophagy and thus significantly downregulated in Rubicon-ablated or aged adipocytes. We propose that age-dependent loss of adipose Rubicon results in excess autophagy, which decreases the levels of SRC-1 and TIF2, contributing to the development of age-associated metabolic disorders.

## Results

**Rubicon deletion in adipocytes causes systemic fat atrophy.** In previous studies, we showed that upregulation of Rubicon, a negative regulator of autophagy[26], is a signature of aging[30]. We hypothesised that Rubicon also accumulates in aged adipocytes, and that its accumulation could decrease autophagic activity in adipose tissue. Surprisingly, we found that adipose tissue in aged mice exhibited a significant decline in levels of Rubicon and the autophagic substrate p62[12] (Fig. 1a), suggesting that autophagic activity in

adipocytes increases with age. Next, to determine the fundamental role of Rubicon in adipose tissue, we genetically deleted Rubicon specifically in adipocytes. To this end, we crossed Adipoq-Cre mice[31] with Rubicon-floxed mice[29] (Supplementary Fig. 1a) to generate Rubicon^{flox/flox}; Adipoq-Cre mice (Rubicon^{ad−/−} mice) and confirmed that Cre-mediated recombination (Supplementary Fig. 1b) and the absence of Rubicon occurred specifically in both the WAT (Supplementary Fig. 1c) and BAT (Supplementary Fig. 1d) of Rubicon^{ad−/−} mice. Rubicon knockout had no significant effect on other ATG protein levels in both adipose tissues (Supplementary Fig. 2a, b). Rubicon^{ad−/−} mice were born in a Mendelian ratio, exhibited no visible abnormalities at birth, and had normal body weight at 5 weeks of age. These observations suggested that Rubicon deletion by Adipoq-Cre does not cause significant abnormality during development. Notably, after 7 weeks of age, the body weight of Rubicon^{ad−/−} mice increased at a lower rate than their control counterparts, and failed to catch up even at 21 weeks of age on a normal chow diet (NCD) (Fig. 1b, c). Consistent with this, we observed a significant decrease in the organ weights of epididymal (eWAT), mesenteric (mWAT) and inguinal WAT (iWAT), as well as interscapular BAT (iBAT), but not liver, in Rubicon^{ad−/−} mice (Fig. 1d, e). Haematoxylin and eosin (H&E) staining of the tissue sections revealed that Rubicon deficiency caused a reduction in adipose cell size in eWAT (Fig. 1f, g) and iBAT (Fig. 1h). These facts suggest that Rubicon maintains the size of adipocytes. Remarkably, these characteristics of Rubicon knockout were not observed in mice also harbouring a knockout of Atg5, a gene essential for autophagy (Fig. 1b, d, f–h). We confirmed that the reduction of Rubicon and ATG12–ATG5 complex in the double knockout mice was comparable with that in the single knockout mice (Supplementary Fig. 1c, d). We also found that Rubicon^{ad−/−} mice exhibited a significant decrease in the levels of two autophagic substrates, LC3-II and p62, but not LC3-II/I ratio, in eWAT (Supplementary Fig. 2c) and the iBAT (Supplementary Fig. 2d). These decreases were completely abolished by additional Atg5 deletion (Supplementary Fig. 2c, d). Importantly, An ex vivo autophagic flux assay[20,21] revealed an increase in autophagic activity in the eWAT (Supplementary Fig. 2e), the iWAT (Supplementary Fig. 2f) and the iBAT (Supplementary Fig. 2g) in Rubicon^{ad−/−} mice. These data collectively indicate that systemic fat atrophy in Rubicon^{ad−/−} mice is caused by inadequate upregulation of autophagy. Rubicon participates positively in LC3-associated phagocytosis (LAP);[32] however, the phenotypes of Rubicon^{ad−/−} mice must be independent of LAP because their phenotypes were abolished by additional knockout of Atg5, which is also required for this process[32].

Notably, neither deletion of Rubicon nor Atg5 had any effect on dietary intake (Supplementary Fig. 3a), oxygen consumption (Supplementary Fig. 3b), or carbon dioxide production (Supplementary Fig. 3c), indicating that the observed phenotypes in Rubicon^{ad−/−} mice did not arise from changes in food intake or energy consumption. Moreover, neither Rubicon nor Atg5 knockout had an effect on the respiratory quotient (RQ) (Supplementary Fig. 3d), excluding the possibility that the fat atrophy in Rubicon^{ad−/−} mice was caused by switching between glucose and fat utilisation in vivo. Taken together, our findings show that the loss of Rubicon in adipocytes causes systemic atrophy without affecting energy balance.

**Rubicon deletion in adipocytes leads to metabolic disorders.** To determine whether systemic fat atrophy in Rubicon^{ad−/−} mice is associated with systemic changes in metabolism, we measured a series of indicators of metabolic disorder. Strikingly, we found that plasma triglyceride (Fig. 2a) and cholesterol (Fig. 2b) levels were significantly higher in Rubicon^{ad−/−} mice than in control

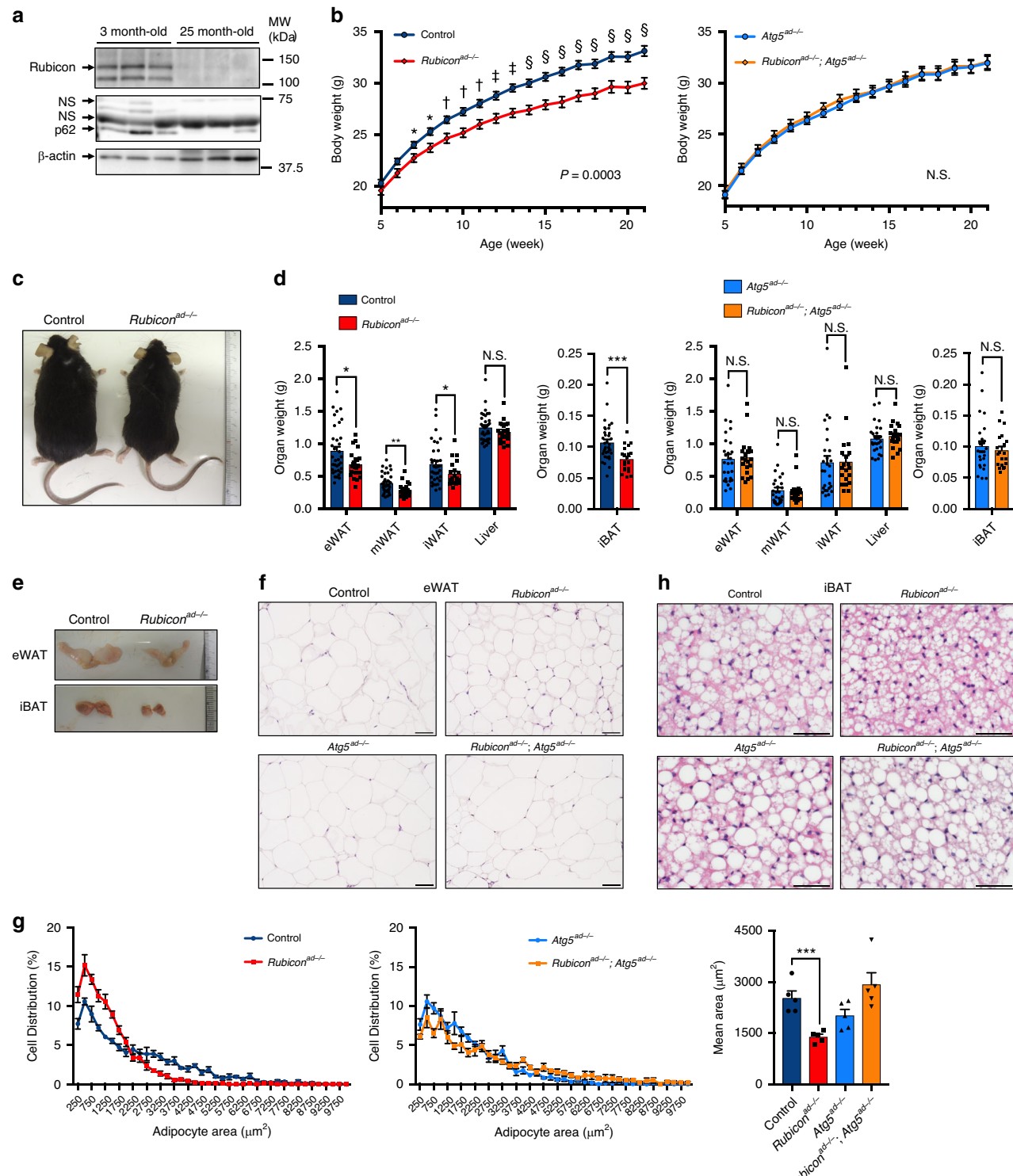

**Fig. 1 Loss of *Rubicon* in adipocytes causes a reduction in body weight and adipose tissue mass. a** Immunoblotting to detect Rubicon and p62 in the eWAT depots of 3- or 25-month-old wild-type mice on an NCD. *n* = 3 mice. **b** Body weight chart for mice of the indicated genotypes on an NCD. Intraperitoneal glucose tolerance test (IPGTT) (Fig. 2c) and intraperitoneal insulin tolerance test (IPITT) were performed at 17 and 19 weeks, respectively. Control, *n* = 35; *Rubicon*^ad−/−, *n* = 21; *Atg5*^ad−/−, *n* = 27; *Rubicon*^ad−/−; *Atg5*^ad−/−, *n* = 20. **c** Representative image of a 21-week-old control and *Rubicon*^ad−/− mouse on an NCD. **d** Organ weight of eWAT, mWAT, iWAT, iBAT and liver from 21-week-old mice of the indicated genotypes, described in (**b**). Control, *n* = 35; *Rubicon*^ad−/−, *n* = 21; *Atg5*^ad−/−, *n* = 27; *Rubicon*^ad−/−; *Atg5*^ad−/−, *n* = 20. **e** Representative images of fat pads from a 21-week-old control and *Rubicon*^ad−/− mouse on an NCD. **f, h** Representative images of H&E staining of eWAT (**f**) and iBAT (**h**) sections from mice of the indicated genotypes on an NCD. Scale bars, 50 μm. *n* = 5 mice with similar results. **g** Distribution of adipocyte area in the eWAT sections in **f**. *n* = 5 mice. Quantification of mean adipocyte area is shown in the graphs at right. Error bars indicate means ± SEM. Data were analysed by two-tailed Student's *t* test (**d**, **g**) or two-way repeated-measures ANOVA followed by Fisher's LSD test (**b**). *P* value from left to right: 0.0003, 0.7257 (**b**), 0.0258, 0.0067, 0.0494, 0.2337, 0.0005, 0.8217, 0.8418, 0.9407, 0.5023, 0.5205 (**d**), 0.0008 (**g**). *P < 0.05; ** or †P < 0.01; *** or ‡P < 0.001; §P < 0.0001. N.S. not significant.

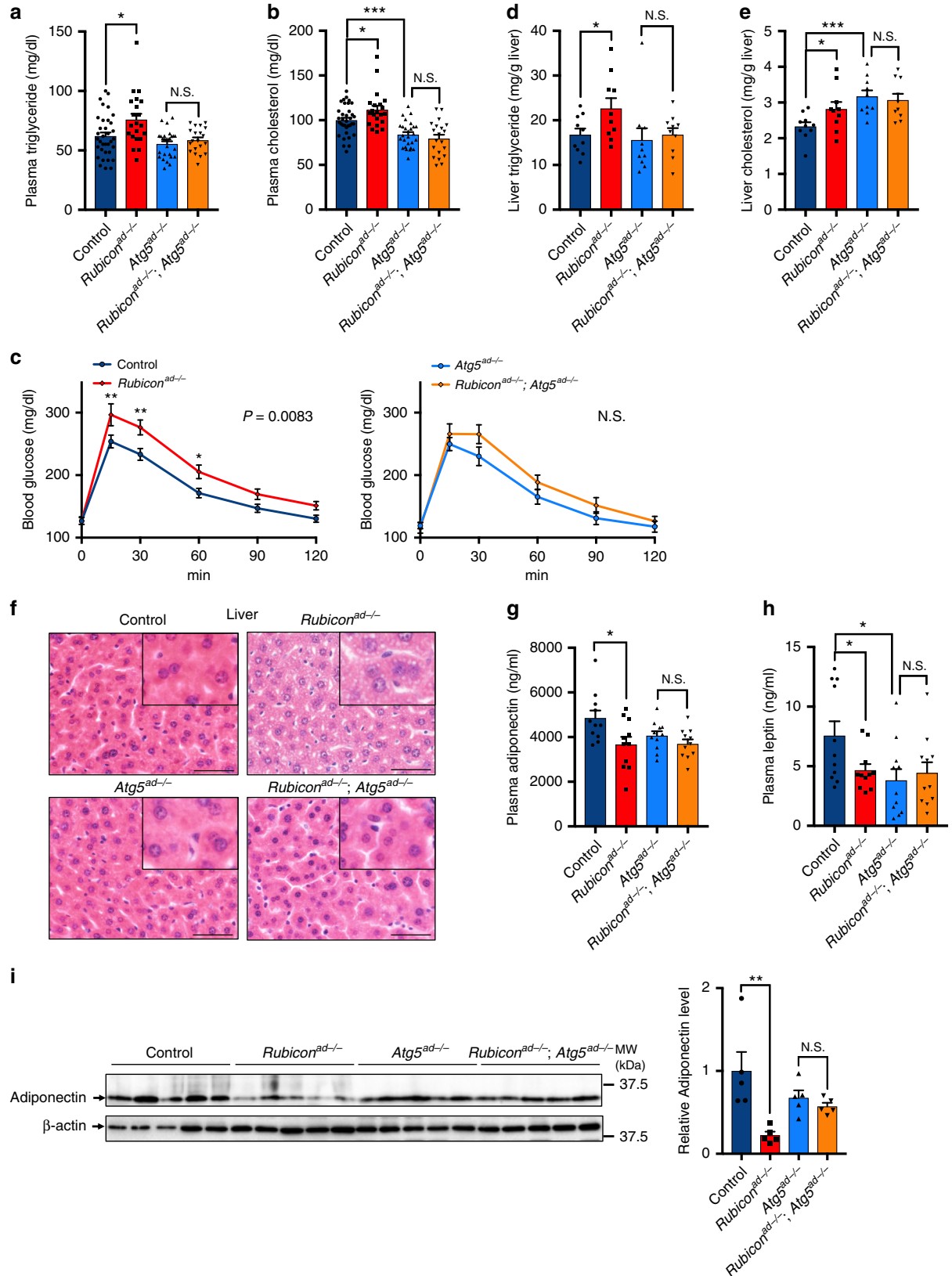

mice. In addition to lipid metabolic disorders, *Rubicon^ad−/−* mice exhibited glucose intolerance, which was less significant in an autophagy-deficient background (Fig. 2c), strongly suggesting that Rubicon in adipocytes plays a crucial role in systemic metabolism by regulating proper autophagic activity. Consistent with this idea, we found that *Rubicon^ad−/−* mice exhibited

hepatic lipid accumulation. *Rubicon* knockout in adipocytes increased hepatic triglyceride (Fig. 2d) and cholesterol content (Fig. 2e). Histological analysis revealed that *Rubicon* deficiency in adipocytes caused abnormal LD formation in liver (Fig. 2f). We assumed that metabolic disorders in *Rubicon^ad−/−* mice stemmed from a decline not only in the lipid storage function, but also in

**Fig. 2 Loss of *Rubicon* in adipocytes disrupts lipid and glucose homoeostasis.** Plasma triglyceride (**a**) and cholesterol (**b**) levels in 21-week-old mice of the indicated genotypes on an NCD. Control, $n = 35$; *Rubicon*$^{ad−/−}$, $n = 21$; *Atg5*$^{ad−/−}$, $n = 27$; *Rubicon*$^{ad−/−}$; *Atg5*$^{ad−/−}$, $n = 20$. **c** Glucose tolerance test in mice of the indicated genotypes on an NCD. Whole-body glucose levels were measured at 15, 30, 60, 90 and 120 min. Control, $n = 35$; *Rubicon*$^{ad−/−}$, $n = 21$; *Atg5*$^{ad−/−}$, $n = 27$; *Rubicon*$^{ad−/−}$; *Atg5*$^{ad−/−}$, $n = 20$. Liver triglyceride (**d**) and cholesterol (**e**) levels in 21-week-old mice of the indicated genotypes on an NCD. $n = 10$ mice. **f** Representative images of H&E staining of liver sections from mice of the indicated genotypes on an NCD. Scale bars, 50 μm. $n = 5$ mice with similar results. Plasma adiponectin (**g**) and leptin (**h**) levels in 21-week-old mice of the indicated genotypes on an NCD. $n = 11$ mice. **i** Immunoblotting to detect adiponectin in the eWAT depots of 21-week-old mice of the indicated genotypes on an NCD. $n = 5$ mice. Quantification of adiponectin levels is shown in the graphs at right. Error bars indicate means ± SEM. Data were analysed by two-tailed Student's $t$ test (**a**, **b**, **d**, **e**, **g**, **h**), one-way ANOVA followed by Tukey's test (**i**), or two-way repeated-measures ANOVA followed by Fisher's LSD test (**c**). $P$ value from top to bottom and left to right: 0.0121, 0.3424 (**a**), 0.0002, 0.0229, 0.3537 (**b**), 0.0083, 0.2176 (**c**), 0.0410, 0.6983 (**d**), 0.0006, 0.0459, 0.6643 (**e**), 0.0215, 0.2115 (**g**), 0.0231, 0.0380, 0.6371 (**h**), 0.0026, 0.9369 (**i**). *$P < 0.05$; **$P < 0.01$; ***$P < 0.001$. N.S. not significant.

the endocrine function. Indeed, we found that the *Rubicon*$^{ad−/−}$ mice had significantly lower levels of plasma adiponectin (Fig. 2g) and leptin (Fig. 2h), which are adipocyte-derived hormones maintaining glucose and lipid metabolism *in vivo*[33–35], whereas *Rubicon* deletion in the *Atg5*-knockout background caused no detectable change in these protein levels (Fig. 2g, h). Western blot of eWAT lysates revealed that adiponectin was markedly reduced in *Rubicon*$^{ad−/−}$ mice (Fig. 2i). This reduction was cancelled in *Rubicon*$^{ad−/−}$; *Atg5*$^{ad−/−}$ mice (Fig. 2i), indicating that *Rubicon* deletion in adipocytes decreases adiponectin production in an autophagy-dependent fashion. We also found that *Atg5*$^{ad−/−}$ mice showed a reduction in plasma cholesterol (Fig. 2b) and leptin (Fig. 2h), and an increase in hepatic cholesterol (Fig. 2e). These results suggest that an autophagy-dependent or -independent role of ATG5 in adipocytes could be important for leptin secretion or cholesterol metabolism.

Our results described above indicated that the loss of Rubicon in adipocytes inappropriately upregulates autophagy, resulting in metabolic disorders. To investigate the role of adipose Rubicon in obesity, we fed *Rubicon*$^{ad−/−}$ mice a high-fat diet (HFD). Notably, *Rubicon*$^{ad−/−}$ mice weighed less than controls fed an HFD (Supplementary Fig. 4a), and adipose-specific *Rubicon*-knockout mice fed an HFD exhibited reduced adipose weight (Supplementary Fig. 4b) and adipose cell size (Supplementary Fig. 4c–e), but an increase in glucose intolerance (Supplementary Fig. 4f) and hepatic steatosis (Supplementary Fig. 4g, h). Obese *Rubicon*$^{ad−/−}$ mice exhibited a reduction in plasma adiponectin (Supplementary Fig. 4i), but not leptin (Supplementary Fig. 4j). These phenotypes of obese *Rubicon*$^{ad−/−}$ mice were not seen in the *Atg5*-deleted background (Supplementary Fig. 4a–j). Together, these results indicate that Rubicon in adipocytes maintains glucose and lipid metabolism by modulating autophagy independently of diet. Interestingly, HFD-fed *Atg5*$^{ad−/−}$ mice exhibited a reduction in visceral eWAT and an increase in subcutaneous iWAT and iBAT, compared with the control counterparts (Supplementary Fig. 4b). This fat maldistribution was associated with a reduction in plasma adiponectin (Supplementary Fig. 4i), and an increase in obesity-induced glucose intolerance (Supplementary Fig. 4f) and hepatic triglyceride accumulation (Supplementary Fig. 4h). The *Atg5* deficiency in adipocytes also promoted formation of the HFD-induced crown-like structure (CLS) in the eWAT (Supplementary Figs. 4c and 8d), consistent with a previous report demonstrating the role of ATG3 and ATG16L1 in adipose tissue inflammation during obesity[36]. These results suggest that ATG conjugation system, which includes ATG3, ATG5 and ATG16L1, in adipocytes could prevent obesity-related metabolic disorders.

**Autophagic activity in adipose tissue increases with age**. To further investigate the role of autophagy in adipocytes during the aging process, we examined Rubicon protein levels in adipose tissue from 12- or 18-month-old mice, and found that the levels of Rubicon and p62 were significantly reduced at the age of 12 months (Fig. 3a). Rubicon mRNA levels also decreased with age, whereas p62 mRNA levels did not (Fig. 3b), suggesting that Rubicon is downregulated with age at the transcriptional level, followed by upregulation of autophagy that is marked by a decrease in an autophagic substrate p62 at a protein level. Importantly, an ex vivo autophagic flux assay using LC3-II[20,21] revealed that autophagic activity in adipose tissue increased with age (Fig. 3c). At the age of 25 months, a decline in transcripts of p62 in eWAT became evident as well (Supplementary Fig. 5a), suggesting a possible involvement of transcriptional regulation in age-associated changes of adipose p62.

Next, to clarify the role of adipose Rubicon in the aging process, we allowed *Rubicon*$^{ad−/−}$ mice to age until they were 12 or 18 months old. Remarkably, the aged *Rubicon*$^{ad−/−}$ mice exhibited reductions in body weight (Fig. 3d), adipose tissue weight (Fig. 3e and Supplementary Fig. 5b–e) and adipose cell size (Fig. 3f–h and Supplementary Fig. 5f) at both ages relative to their control counterparts. Plasma triglyceride (Supplementary Fig. 5g) and cholesterol levels (Supplementary Fig. 5h) were not significantly increased with age in mice, but were higher in aged *Rubicon*$^{ad−/−}$ mice than in aged control mice. Like the increase in autophagic activity (Fig. 3c), the phenotypes of aged *Rubicon*$^{ad−/−}$ mice were abolished by additional deletion of *Atg5* (Fig. 3d–h and Supplementary Fig. 5b–h). Collectively, these data indicate that Rubicon in adipocytes prevents excess autophagy, thereby maintaining metabolic homoeostasis over the aging process. More importantly, we found that hepatic triglyceride (Fig. 3i) and cholesterol levels (Fig. 3j) increased with age, and that these increases were significantly attenuated in aged *Atg5*$^{ad−/−}$ mice. Histological analysis also revealed that the loss of autophagy in adipocytes ameliorated age-associated hepatic steatosis (Fig. 3k). Taken together, our data strongly suggest that excess autophagy due to the loss of Rubicon promotes age-associated metabolic disorders, including liver steatosis.

**Rubicon is upregulated during adipogenesis**. To determine the molecular mechanism by which Rubicon maintains adipocyte homoeostasis, we monitored changes in autophagic activity and *Rubicon* expression along with adipogenesis. We found that during adipogenesis, 3T3-L1 cells exhibited a gradual decrease in autophagic flux, as determined based on the degradation of autophagic substrates LC3-II and p62 (Fig. 4a). This was associated with an increase in Rubicon protein (Fig. 4b) and mRNA levels (Fig. 4c). Because *Rubicon* knockdown, which results in upregulation of autophagy (Fig. 4d), dramatically decreased Adiponectin protein level in differentiated 3T3-L1 cells (Fig. 4e), the increase in Rubicon must be crucial for adipogenesis. Indeed, we found that *Rubicon* knockdown decreased the expression of a wide range of adipogenic genes, including *Adipoq*, *Lep*, *Fabp4*,

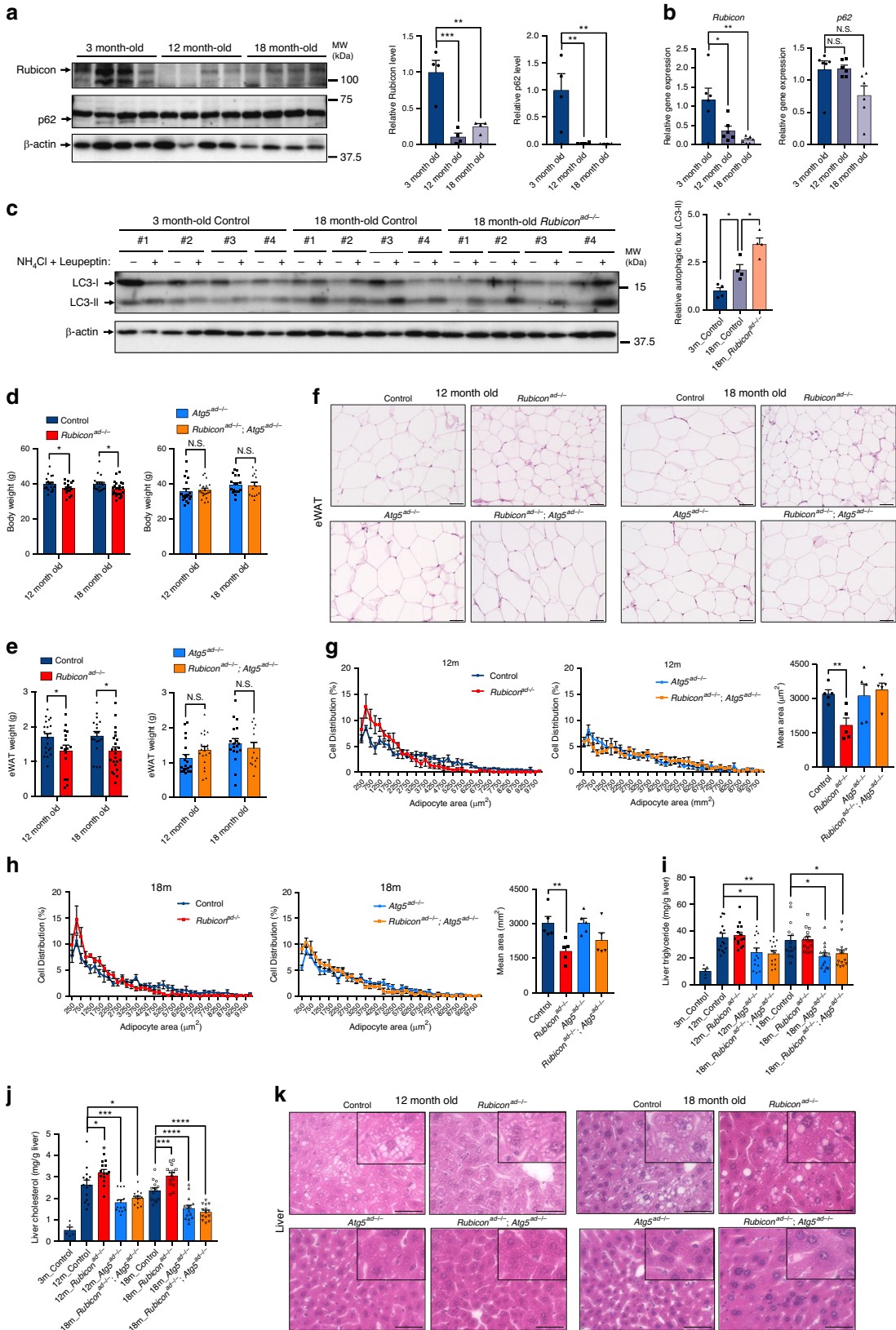

*Cd36*, *Glut4*, *Acaca* and *Fasn* (Fig. 4f). Consistent with this, we found that *Rubicon* deletion in adipocytes in vivo caused a decrease in adipogenic gene expression that was again reverted by additional *Atg5* knockout (Fig. 4g).

To further investigate lipid metabolism pathways in *Rubicon*[ad−/−] mice, we examined the expression of Perilipin1 and Perilipin2, which participate in lipolysis or LD stabilisation[37]. Perilipin1 and Perilipin2 were downregulated at the transcriptional level in

**Fig. 3 Excess autophagy in adipocytes promotes age-related hepatic steatosis. a** Immunoblotting to detect Rubicon and p62 in eWAT depots of 3-, 12- and 18-month-old control mice. $n = 4$ mice. **b** Relative mRNA expression of *Rubicon* and *p62* in eWAT depots of 3-, 12- and 18-month-old control mice. $n = 6$ mice. **c** Ex vivo autophagic flux assay based on LC3-II degradation in the eWAT depots of mice of the indicated ages and genotypes. $n = 4$ mice. Body weight (**d**) and eWAT weight (**e**) of mice of the indicated ages and genotypes. 12-month control, $n = 21$; 12-month *Rubicon*$^{ad-/-}$, $n = 16$; 12-month *Atg5*$^{ad-/-}$, $n = 19$; 12-month *Rubicon*$^{ad-/-}$;*Atg5*$^{ad-/-}$, $n = 20$; 18-month control; $n = 18$; 18-month *Rubicon*$^{ad-/-}$, $n = 23$; 18-month *Atg5*$^{ad-/-}$, $n = 18$; 18-month *Rubicon*$^{ad-/-}$;*Atg5*$^{ad-/-}$, $n = 13$. Representative images of H&E staining of eWAT (**f**) and liver (**k**) sections from mice of the indicated ages and genotypes. Scale bars, 50 μm. $n = 5$ mice with similar results. Distribution of adipocyte area in eWAT sections from 12- (**g**) and 18- (**h**) month-old mice in (**f**). $n = 5$ mice. Quantification of mean adipocyte area is shown in the graphs at right. Liver triglyceride (**i**) and cholesterol (**j**) levels in mice of the indicated ages and genotypes. $n = 14$ mice, except for 3-month-old mice ($n = 5$). Aged mice were maintained on an NCD. Quantification data are shown in the graphs at the right of each blot. Error bars indicate means ± SEM. Data were analysed by two-tailed Student's *t* test (**d**, **e**, **g**, **h**), one-way ANOVA followed by Tukey's test (**a–c**) or Dunnett's test (**i**, **j**). $P$ value from top to bottom and left to right: 0.0014, 0.0004, 0.0079, 0.0081 (**a**), 0.0039, 0.0237, 0.0753, 0.9927 (**b**), 0.0365, 0.0136 (**c**), 0.0432, 0.0293, 0.6662, 0.8681 (**d**), 0.0362, 0.0157, 0.1353, 0.5337 (**e**), 0.0047 (**g**), 0.0093 (**h**), 0.0077, 0.0148, 0.0430, 0.0125 (**i**), 0.0145, 0.0009, 0.0242, <0.0001, <0.0001, 0.0007 (**j**). *$P < 0.05$; **$P < 0.01$; ***$P < 0.001$; ****$P < 0.0001$. N.S. not significant.

*Rubicon*$^{ad-/-}$ mice (Supplementary Fig. 6a, b), possibly causing the reduction in LD size. We also found that gene expression of lipoprotein lipase (LPL), which hydrolyses triglycerides in chylomicrons or VLDL[38], was significantly decreased in the adipose tissue of *Rubicon*$^{ad-/-}$ mice (Supplementary Fig. 6c). In addition, expression of ABCA1 and ABCG1, which participate in cholesterol efflux[38], was elevated in *Rubicon*$^{ad-/-}$ mice (Supplementary Fig. 6c), whereas levels of the cholesterol receptor LDL-R and SR-B1[38] were not changed (Supplementary Fig. 6c). Previous reports showed that knockout of *Lpl* increases both plasma triglyceride and cholesterol[39], and that adipose-specific knockout of *Abca1* decreases plasma cholesterol[40]. Therefore, the reduction of LPL and the increases in ABC proteins could mediate the lipid metabolic disorders in *Rubicon*$^{ad-/-}$ mice. *Rubicon* deletion in adipocytes did not increase the expression of lipogenic genes in the liver (Supplementary Fig. 6d), supporting the idea that the lipids in adipocytes could be transferred into the liver in *Rubicon*$^{ad-/-}$ mice. Moreover, we investigated whether *Rubicon* knockout induces lipolysis in adipocytes, and found that *Rubicon* deletion had no impact on characteristic features of BAT (Supplementary Fig. 6e). The lipolytic genes *Atgl* and *Hsl*[41] in WAT tended to be downregulated or at least not changed (Supplementary Fig. 6f). *Rubicon* deletion in adipocytes did not increase the plasma levels of non-esterified fatty acid (NEFA), a product of lipolysis, under either fed or fasted conditions (Supplementary Fig. 6g). Notably, ex vivo lipolysis assay showed that catecholamine-induced release of glycerol was not significantly increased in eWAT (Supplementary Fig. 6h), iWAT (Supplementary Fig. 6i) or iBAT (Supplementary Fig. 6j) from *Rubicon*$^{ad-/-}$ mice. That of NEFA was not also increased in *Rubicon*$^{ad-/-}$ mice (Supplementary Fig. 6k–m). These data collectively suggest that the loss of *Rubicon* does not enhance lipolysis in adipocytes. Notably in this regard, the reduction in adipogenic gene expressions was also seen in younger *Rubicon*$^{ad-/-}$ mice (Supplementary Fig. 6n), suggesting that Rubicon is needed for early adipocyte development.

Considering a role of mitophagy[42], mitochondria-targeted autophagy, in adipogenesis[15,16], we tested whether any possible changes in mitophagy due to the loss of Rubicon in adipocytes contributes to the phenotype shown above. We assessed the mitochondrial respiration in *Rubicon*-knockdown adipocytes using tetramethylrhodamine methyl ester (TMRM), an indicator of the mitochondrial membrane potential. We found that the loss of Rubicon did not cause any change in the mitochondrial membrane potential while an ionophore valinomycin remarkably decreased the membrane potential as a positive control (Supplementary Fig. 7a). For assessment of Parkin-dependent mitophagy, we treated the cells with valinomycin to see the recruitment of Parkin onto the mitochondria, and found that Parkin was recruited to the damaged mitochrondria in *Rubicon*-knockdown cells at a level comparable to control (Supplementary Fig. 7b). Consistently, *Rubicon* knockdown in adipocytes did not

affect the interaction between LC3-II and p62 that works as a potential receptor for mitophagy (Supplementary Fig. 7c). The degradation rate of outer and inner mitochondrial proteins was not significantly changed in *Rubicon*-knockdown cells (Supplementary Fig. 7d). In addition, the protein levels of Parkin and PINK1 were not affected in the *Rubicon*-knockdown cells (Supplementary Fig. 7d) or the WAT (Supplementary Fig. 2a) and the BAT (Supplementary Fig. 2b) of *Rubicon*$^{ad-/-}$ mice. These data collectively indicate that the downregulation of Rubicon in adipocytes does not cause any significant change in mitophagy. Furthermore, ex vivo culture of adipose tissue with a lysosomal inhibitor did not increase the protein level of LD-resident Perilipin1, Perilipin2 and mitochondrial Complex III subunit Core 1, but did increase LC3-II (Supplementary Fig. 7e), suggesting that lipophagy[42] or mitophagy in adipocytes works only at an undetectable level and may play a relatively minor role in the absence of exogenous stimuli.

Cell death or inflammation often leads to a reduction in adipogenic gene expression in adipose tissue. Moreover, excess autophagy is closely involved with cell death[43,44]; thus, upregulation of autophagy by *Rubicon* knockout could affect cell proliferation or cell death. We performed immunohistochemistry for PCNA to assess cell proliferation in *Rubicon*$^{ad-/-}$ mice. Although PCNA-positive cells were observed in the epididymis of control mice, PCNA-positive adipocytes were not detected in eWAT in either control or *Rubicon*$^{ad-/-}$ mice (Supplementary Fig. 8a). This is consistent with the idea that adipocytes are non-proliferative, and that *Rubicon* knockout does not affect adipocyte proliferation. Moreover, *Rubicon* knockout did not increase the level of Cleaved Caspase-3 in adipose tissue (Supplementary Fig. 8b). TUNEL-positive adipocytes were observed in a DNase I-treated sample, used as a positive control, but not in untreated samples from control or *Rubicon*$^{ad-/-}$ mice (Supplementary Fig. 8c). CLSs, dead adipocytes surrounded by macrophages[45], were not detected in the eWAT in either control or *Rubicon*$^{ad-/-}$ mice (Fig. 1f, Supplementary Fig. 8d), whereas they were present in HFD-fed mice (Supplementary Figs. 4c and 8d). These data indicate that adipocyte death is not induced in *Rubicon*$^{ad-/-}$ mice. To determine whether systemic inflammation is induced in *Rubicon*$^{ad-/-}$ mice, we measured the plasma IL-6 or TNFα level, and found that those in *Rubicon*$^{ad-/-}$ mice were comparable to those in control mice (Supplementary Fig. 9a, b). In addition, the gene expression of pro-inflammatory cytokines or macrophage markers was not significantly elevated in eWAT (Supplementary Fig. 9c, d) or iBAT (Supplementary Fig. 9e, f) of *Rubicon*$^{ad-/-}$ mice. Consistently, flow cytometry analysis showed that the rate of CD45$^+$ hematopoietic cells was not increased in the stromal vascular fraction (SVF) from the eWAT of *Rubicon*$^{ad-/-}$ mice (Supplementary Fig. 9g). The rate of CD11b$^+$ F4/80$^+$ macrophage in the CD45$^+$ hematopoietic cells was not increased in

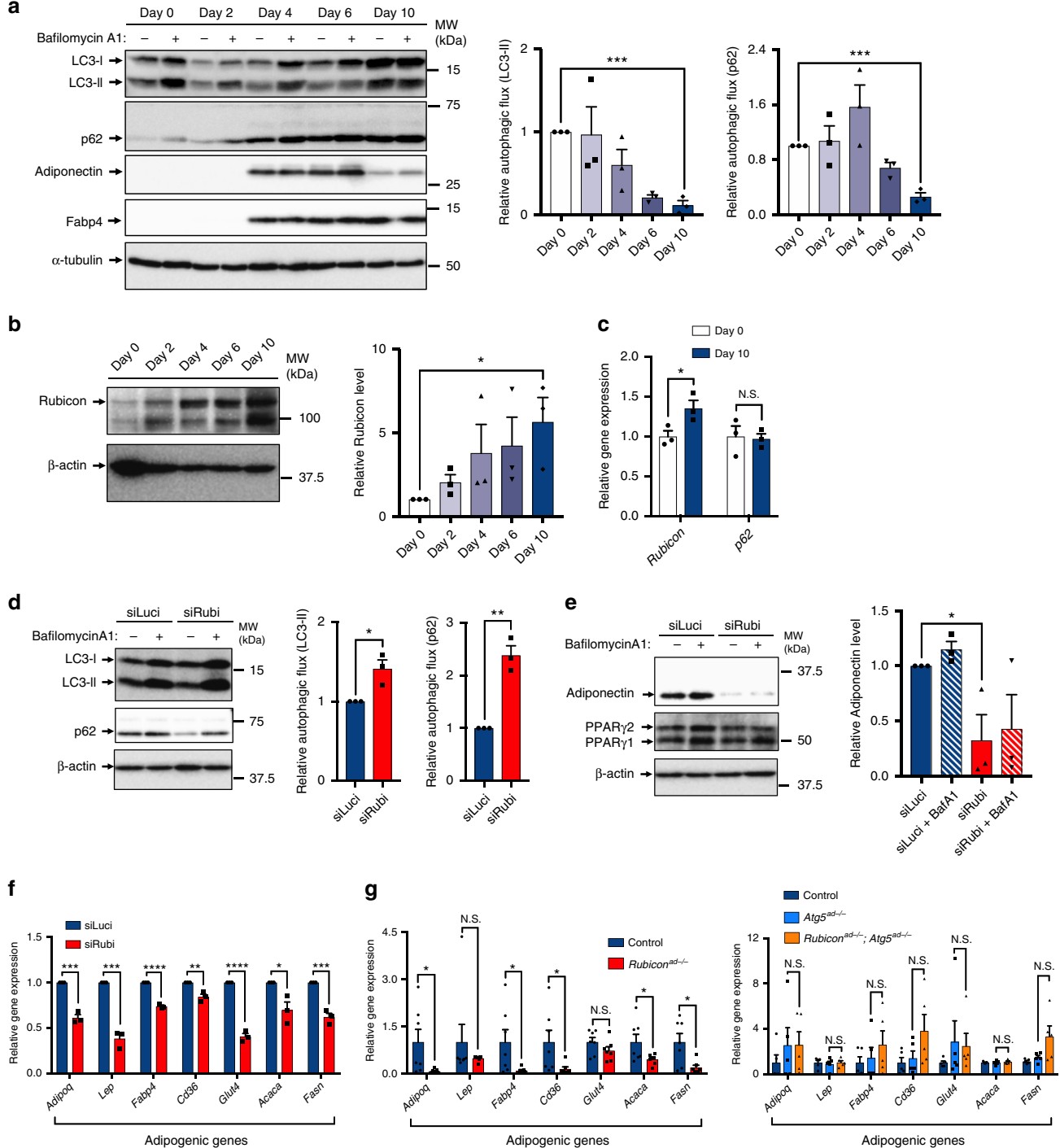

**Fig. 4 Upregulation of Rubicon during adipogenesis ensures adipogenic gene expression. a** Autophagic flux assay based on LC3-II and p62 degradation in 3T3-L1 cells on days 0, 2, 4, 6 and 10 following induction of differentiation. $n = 3$ independent experiments. **b** Immunoblotting to detect Rubicon in 3T3-L1 cells on days 0, 2, 4, 6 and 10 following induction of differentiation. $n = 3$ independent experiments. **c** Relative mRNA expression of *Rubicon* and *p62* in 3T3-L1 cells on days 0 and 10 following induction of differentiation. $n = 3$. **d** Autophagic flux assay based on LC3-II and p62 degradation in *Luciferase*- or *Rubicon*-knockdown 3T3-L1 cells. Knockdown was carried out for 48 h starting on day 8 following induction of differentiation. $n = 3$ independent experiments. **e** Immunoblotting to detect adiponectin in *Luciferase*- or *Rubicon*-knockdown 3T3-L1 cells, treated with or without 125 nM Baf A1 for 2 h. Knockdown for 48 h was carried out on day 8 following induction of differentiation. $n = 3$ independent experiments. **f** Relative mRNA expression of adipogenic genes in *Luciferase*- or *Rubicon*-knockdown 3T3-L1 cells. Knockdown was carried out for 48 h starting on day 8 following induction of differentiation. $n = 3$ independent experiments. **g** Relative mRNA expression of adipogenic genes in eWAT depots of 21-week-old mice of the indicated genotypes on an NCD. $n = 7$ mice (left); $n = 5$ mice (right). Quantification data are shown in the graphs at the right of each blot. Error bars indicate means ± SEM. Data were analysed by two-tailed Student's $t$ test (**a–g**). $P$ value from left to right: 0.0001, 0.0002 (**a**), 0.0335 (**b**), 0.0444, 0.8445 (**c**), 0.0210, 0.0016 (**d**), 0.0443 (**e**), 0.0004, 0.0004, <0.0001, 0.0070, <0.0001, 0.0271, 0.0008 (**f**), 0.0393, 0.3729, 0.0400, 0.0417, 0.1710, 0.0477, 0.0129, 0.9921, 0.6594, 0.4752, 0.1719, 0.8678, 0.5768, 0.0913 (**g**). *$P < 0.05$; **$P < 0.01$; ***$P < 0.001$; ****$P < 0.0001$. N.S. not significant.

*Rubicon*$^{ad-/-}$ mice (Supplementary Fig. 9h). Histological analysis also revealed that inflammatory cells were not seen in the eWAT (Fig. 1f) or the iBAT (Fig. 1h) in *Rubicon*$^{ad-/-}$ mice. These results suggest that inflammatory cell infiltration does not occur in adipose tissue of *Rubicon*$^{ad-/-}$ mice. Based on these findings, we conclude that cell death and inflammation are not major contributors to adipocyte dysfunction in *Rubicon*$^{ad-/-}$ mice.

**Activation of PPARγ can restore *Rubicon*-ablated adipocytes.** Our data suggest that downregulation of Rubicon causes adipocyte dysfunction, which could stem from a reduction in a wide range of adipogenic gene expressions. Because PPARγ is a master transcription factor involved in adipogenesis[46], and *Adipoq*-Cre-mediated knockout of *Pparg* causes a phenotype that is similar to, but much stronger than, that of *Rubicon* knockout[47], we hypothesised that PPARγ activity is reduced in *Rubicon*$^{ad-/-}$ mice. To test this hypothesis, we treated *Rubicon*-depleted cells with thiazolidinedione (TZD), an exogenous ligand of PPARγ[48]. Strikingly, TZD treatment restored the reduction in Adiponectin protein level (Fig. 5a), adipogenic gene expression (Fig. 5b) and triglyceride content (Fig. 5c) caused by *Rubicon* depletion. Consistent with this, Oil Red O (ORO) staining revealed that *Rubicon* depletion decreased LDs (Fig. 5d), whereas no such reduction was seen under TZD treatment (Fig. 5d). Furthermore, we treated *Rubicon*$^{ad-/-}$ mice with TZD, and found that TZD treatment in vivo almost completely rescued the reduction in plasma adiponectin (Fig. 5e) and leptin (Fig. 5f) levels, as well as adipose mass (Fig. 5g). Our findings suggest that activation of PPARγ reverses adipocyte dysfunction caused by loss of Rubicon.

**SRC-1 and TIF2 are degraded by autophagy in adipocytes.** The results described above indicate that upregulation of autophagy by loss of Rubicon causes adipocyte dysfunction, which could cause PPARγ activity to be suppressed. Therefore, we assumed that autophagy degrades the PPARγ coactivator in adipocytes in *Rubicon*$^{ad-/-}$ mice. Previous reports showed that both SRC-1/NCoA1 and TIF2/SRC-2/NCoA2 work as coactivators of PPARγ[49], and that SRC-1 and TIF2 are localised both in the cytosol and nucleus[50,51]. Importantly, a protein in the same family, NCoA4, is an autophagic substrate[52]. These reports prompted us to investigate whether SRC-1 and/or TIF2 are degraded by autophagy in *Rubicon*-ablated adipocytes. Remarkably, *Rubicon* depletion decreased the levels of both SRC-1 and TIF2 proteins in differentiated 3T3-L1 cells, whereas the lysosomal inhibitor increased the levels of both proteins (Fig. 6a). Also, in mice, SRC-1 and TIF2 levels were increased in an ex vivo culture with lysosomal inhibitor (Fig. 6b). *Rubicon*$^{ad-/-}$ mice exhibited decreases in SRC-1 and TIF2 (Fig. 6c), but these reductions were abolished in the *Atg5*-knockout background (Fig. 6d). Collectively, these data indicate that SRC-1 and TIF2 are degraded by autophagy, and thus downregulated, in adipocytes of *Rubicon*$^{ad-/-}$ mice. We also found that SRC-1 and TIF2 were dramatically decreased in aged adipose tissue, along with the reduction in adiponectin (Fig. 6e, f), suggesting that the reductions in SRC-1 and TIF2 could contribute to age-dependent adipocyte dysfunction. Notably, in addition to aged mice, Rubicon was significantly reduced in fasted mice, concomitant with the reductions in SRC-1 and TIF2 (Fig. 6g). 3T3-L1 adipocytes also showed a starvation-induced decline in Rubicon, SRC-1 and TIF2 (Fig. 6h), followed by a reduction in most of adipogenic gene transcripts (Fig. 6i). These data implicate the physiological system that a reduction of Rubicon in adipocytes actively causes a decline in adipocyte function during fasting.

**SRC-1 and TIF2 interact with GABARAP family proteins.** The LC3-interacting region (LIR)/GABARAP-interacting motif (GIM) is necessary for autophagic degradation of various substrates[53]. Using the iLIR database[54], we found that SRC-1 and TIF2 have evolutionarily conserved LIR/GIM-like motifs (Fig. 7a), which are comparable to the ATG8-interacting motif in yeast ATG3[55], suggesting that their autophagic degradation is mediated by binding to LC3 and/or GABARAP. To test this possibility, we generated mutant constructs in which the tryptophan in the potential LIR/GIM was replaced with alanine, and performed immunoprecipitation of FLAG-tagged SRC-1 or TIF2. Importantly, GABARAP and GATE16, but not LC3, were co-immunoprecipitated with SRC-1 (Fig. 7b), as well as TIF2 (Fig. 7c). The LIR/GIM mutant SRC-1 and TIF2 exhibited a reduction in the interaction with endogenous GABARAP family proteins (Fig. 7b, c). Our findings suggest that the tryptophan in the LIR/GIM of SRC-1 or TIF2 is essential for the interaction with GABARAP or GATE16. Moreover, the LIR/GIM mutant SRC-1 and TIF2 failed to exhibit an increase in the presence of a lysosomal inhibitor (Fig. 7d, e), as well as a reduction upon *Rubicon* depletion (Fig. 7f, g), suggesting that autophagic degradation of SRC-1 and TIF2 by *Rubicon* depletion is mediated by the interaction with GABARAP family proteins. To clarify their contribution to the phenotype in *Rubicon*-knockdown adipocytes, we overexpressed the LIR/GIM mutant SRC-1 and/or TIF2 in *Rubicon*-knockdown cells, and found that both overexpression of the mutant SRC-1 and TIF2 rescued the reduction in adipogenic gene expressions in *Rubicon*-knockdown cells (Fig. 7h). Taken together, these findings indicate that the age-dependent loss of Rubicon causes adipocyte dysfunction, which is mediated by autophagic degradation of both SRC-1 and TIF2, leading to age-associated metabolic disorders.

## Discussion

Here, we showed that downregulation of autophagy by Rubicon is achieved during adipogenesis and is crucial for adipocyte function, and that excess autophagy with the loss of Rubicon in adipocytes occurs in aging, leading to age-associated metabolic disorders. Consistent with these findings, previous studies showed that the insulin receptor (IR)–AKT–mTOR complex 1 (mTORC1) axis[56], which negatively regulates autophagy via phosphorylation of the ULK1 complex[57] and TFEB[58], also maintains adipocyte function. Indeed, recent studies using *Adipoq*-driven Cre have shown that the development or maintenance of adipocyte functions requires each component of the IR–AKT–mTORC1 axis[59–63]. Because this axis promotes lipogenesis by enhancing glucose incorporation, upregulating lipogenic gene sets and increasing glycolysis[64], the strong phenotypes caused by genetic disruption of this axis must arise from multiple factors, including autophagy. However, our results highlight the significance of downregulation of autophagy in the maintenance of adipose tissue homoeostasis. Notably in this regard, the two systems for negatively regulating autophagy are functionally connected: Rubicon is regulated by the mTORC1 pathway via phosphorylation of its binding partner UVRAG[28]. In addition, a previous study showed that mTOR activity in adipose tissue is downregulated with age, but maintained by calorie restriction[65], which extends lifespan in several organisms. Based on these reports, along with the findings in this study, we propose that the age-dependent decline in Rubicon and mTORC1 activity collectively causes adipocyte dysfunction via excess autophagy, leading to age-associated metabolic disorders. Importantly, as in the liver and kidney[30], Rubicon in adipose tissue is also transcriptionally regulated in aging (Fig. 3b). The transcriptional regulator of Rubicon in aged adipocytes remains to be elucidated.

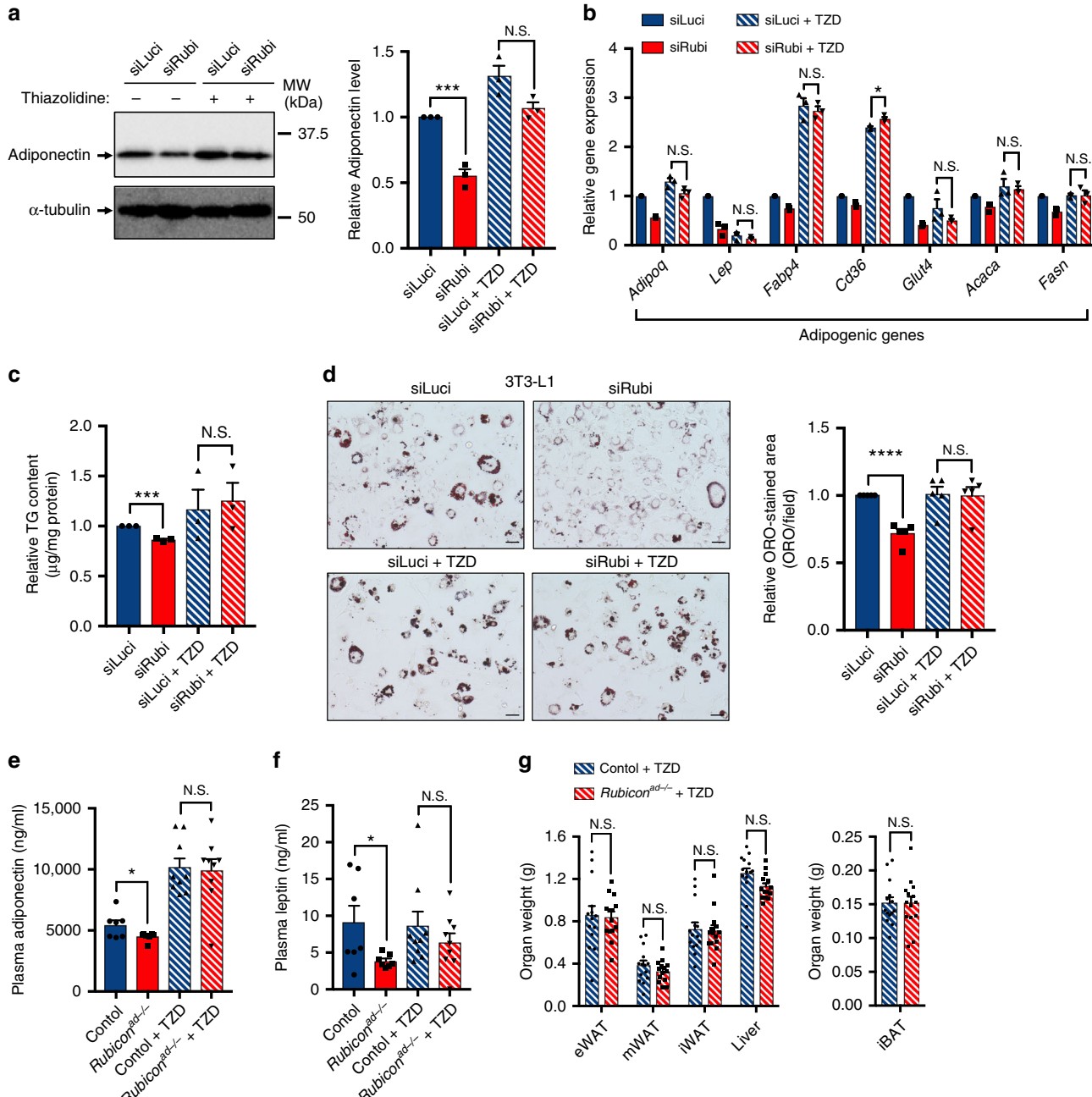

**Fig. 5 PPARγ activation recovers adipocyte dysfunction caused by suppression of Rubicon. a** Immunoblotting to detect adiponectin in *Luciferase*- or *Rubicon*-knockdown 3T3-L1 cells treated with or without 10 μM pioglitazone. Knockdown, with or without TZD treatment, was performed for 48 h starting on day 8 after induction of differentiation. *n* = 3 independent experiments. Quantification of adiponectin levels is shown in the graphs at right. **b** Relative mRNA expression of adipogenic genes in *Luciferase*- or *Rubicon*-knockdown 3T3-L1 cells treated with or without 10 μM pioglitazone. Knockdown, with or without TZD treatment, was performed for 48 h starting on day 8 after induction of differentiation. *n* = 3 independent experiments. **c** Relative triglyceride content in *Luciferase*- or *Rubicon*-knockdown 3T3-L1 cells treated with or without 10 μM pioglitazone. Knockdown, with or without TZD treatment, was performed for 72 h starting on day 8 after induction of differentiation. *n* = 3 independent experiments. **d** Representative images of ORO staining in *Luciferase*- or *Rubicon*-knockdown 3T3-L1 cells treated with or without 10 μM pioglitazone. Knockdown, with or without TZD treatment, was performed for 72 h starting on day 8 after induction of differentiation. Scale bars, 50 μm. *n* = 5 independent experiments. Quantification of relative ORO-stained area is shown in the graphs at right. Plasma adiponectin (**e**) and leptin (**f**) levels in control or *Rubicon^{ad−/−}* mice treated with or without 0.02% pioglitazone for 3 weeks. *n* = 7 mice (without TZD); *n* = 9 mice (with TZD). **g** Organ weight of eWAT, mWAT, iWAT, iBAT and liver from control or *Rubicon^{ad−/−}* mice treated with 0.02% pioglitazone for 3 weeks. *n* = 14 mice. Error bars indicate means ± SEM. Data were analysed by two-tailed Student's *t* test (**a–g**). *P* value from left to right: 0.0009, 0.0650 (**a**), 0.0586, 0.3129, 0.5902, 0.0229, 0.2209, 0.7297, 0.9763 (**b**), 0.0002, 0.7647 (**c**), < 0.0001, 0.9078 (**d**), 0.0450, 0.8353 (**e**), 0.0441, 0.3403 (**f**), 0.8532, 0.2697, 0.8582, 0.0625, 0.9957 (**g**). *P < 0.05; **P < 0.01; ***P < 0.001; ****P < 0.0001. N.S. not significant.

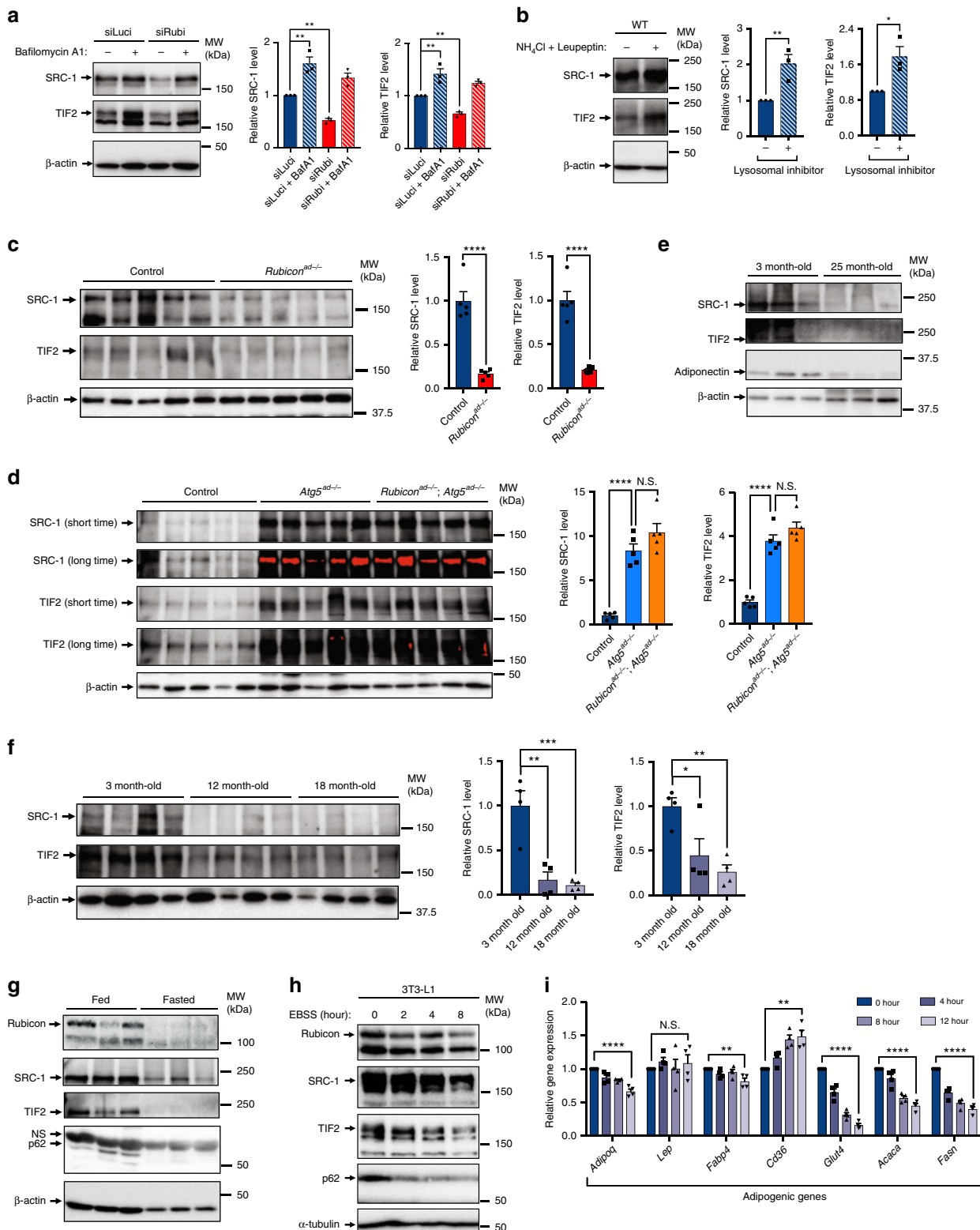

Mechanistically, our findings suggest that excess autophagy by Rubicon suppression in adipocytes leads to an LIR/GIM-dependent reduction in SRC-1 and TIF2, which work as coactivators of PPARγ. It remains to be determined why SRC-1 and TIF2 are degraded by autophagy. One clue regarding the answer to this question is provided by the observation that SRC-1 and TIF2 are significantly decreased in adipose tissue from fasted mice (Fig. 6g), or starved 3T3-L1 adipocytes (Fig. 6h). Therefore, we speculate that SRC-1 and TIF2 in adipocytes are transported to the cytosol and degraded by autophagy in the response to outside energy demand, such as a reduction in insulin signalling, and that these reductions cause a decline in PPARγ activity and adipocyte function, leading to energy supply for other tissues. Furthermore, because overexpression of LIR/GIM mutant SRC-1 and TIF2 did not completely rescue the reduction in adipogenic gene expressions by *Rubicon* knockdown (Fig. 7h), it is possible that the loss

**Fig. 6 SRC-1 and TIF2 are degraded by autophagy, and thus are significantly reduced in *Rubicon*-ablated adipocytes or aged adipocytes.**
**a** Immunoblotting to detect SRC-1 and TIF2 in *Luciferase*- or *Rubicon*-knockdown 3T3-L1 cells treated with or without 125 nM Baf A1 for 8 h. Knockdown was performed for 48 h starting on day 8. $n = 3$ independent experiments. **b** Immunoblotting to detect SRC-1 and TIF2 in wild-type eWAT depots explanted in DMEM, treated with or without 20 mM ammonium chloride and 200 μM leupeptin for 2 h. $n = 3$ independent experiments. **c, d** Immunoblotting to detect SRC-1 and TIF2 in eWAT depots of 21-week-old mice of the indicated genotypes on an NCD. $n = 5$ mice. **e** Immunoblotting to detect the indicated proteins in eWAT depots of 3- or 25-month-old wild-type mice on an NCD. $n = 3$ mice. **f** Immunoblotting to detect SRC-1 and TIF2 in the eWAT depots of 3-, 12-, 18-month-old control mice on an NCD. $n = 4$ mice. **g** Immunoblotting to detect the indicated proteins in eWAT depots of fed or 48-h-fasted 3-month-old wild-type mice. $n = 3$ mice. **h** Immunoblotting to detect the indicated proteins in 3T3-L1 cells. The cells on day 10 were subjected to starvation for the indicated times. $n = 3$ independent experiments. **i** Relative mRNA expression of adipogenic genes in 3T3-L1 cells. The cells on day 10 were subjected to starvation for the indicated times. $n = 4$ independent experiments. Quantification data are shown in the graphs at the right of each blot. Error bars indicate means ± SEM. Data were analysed by two-tailed Student's $t$ test (**b, c, i**), one-way ANOVA followed by Tukey's test (**d, f**) or Dunnett's test (**a**). $P$ value from top to bottom and left to right: 0.0083, 0.0018, 0.0065, 0.0019 (**a**), 0.0163, 0.0256 (**b**), <0.0001, <0.0001 (**c**), <0.0001, 0.1577, <0.0001, 0.1894 (**d**), 0.0008, 0.0013, 0.0077, 0.0359 (**f**), <0.0001, 0.5116, 0.0094, 0.0018, <0.0001, <0.0001, <0.0001 (**i**). \*$P < 0.05$; \*\*$P < 0.01$; \*\*\*$P < 0.001$; \*\*\*\*$P < 0.001$. N.S. not significant.

of *Rubicon* promotes autophagic degradation of other specific component(s) involved in adipogenesis. For example, Caveolin-1, a protein responsible for congenital generalised lipodystrophy[66], was recently shown to be degraded in an autophagy-dependent manner in some cell types[67,68].

In summary, we have revealed that Rubicon plays an essential role in proper maintenance of adipocyte function and systemic metabolic homoeostasis by preventing excess basal autophagy. The absence of Rubicon in adipocytes occurs in aging and leads to metabolic disorders, which are recovered by autophagy deficiency. Thus, we anticipate that inhibition of autophagy in adipocytes could be highly beneficial for human health in an aging society.

## Methods
**Reagents and antibodies.** The following antibodies were used for western blotting at the indicated dilutions: rabbit monoclonal anti-Rubicon (CST, #8465, 1:1000), mouse monoclonal anti-ULK1 (Santa Cruz Biotechnology, sc-33182, 1:2000), mouse monoclonal anti-ATG13 (Sigma-Aldrich, SAB4200100, 1:2000), rabbit polyclonal anti-ATG12 (CST, #2011, 1:2000), rabbit monoclonal anti-ATG16L1 (CST, #8089, 1:2000), mouse monoclonal anti-Beclin 1 (BD, 612112, 1:2000), rabbit polyclonal anti-LC3 (MBL, PM036, 1:2000), mouse monoclonal anti-p62 (MBL, M162-3, 1:2000 for IP samples), rabbit polyclonal anti-p62 (MBL, PM045, 1:5000 for other samples), rat monoclonal anti-Adiponectin (R&D Systems, MAB1119, 1:2000), mouse monoclonal anti-FABP4 (Santa Cruz Biotechnology, sc-271529, 1:2000), mouse monoclonal anti-PPARγ (Santa Cruz Biotechnology, sc-7273, 1:2000), rabbit monoclonal anti-SRC-1 (CST, #2191, 1:2000), rabbit polyclonal anti-TIF2 (Bethyl Laboratories, A300-346A, 1:2000), rabbit monoclonal anti-Perilipin1 (CST, #9349, 1:2000), rabbit polyclonal anti-Perilipin2 (Abcam, ab52356, 1:2000), mouse monoclonal anti-Complex III subunit Core 1 (Invitrogen, #459140, 1:2000), rabbit polyclonal anti-Tomm20 (Santa Cruz, sc-11415, 1:2000), rabbit polyclonal anti-Parkin (CST, #2132, 1:2000), rabbit polyclonal anti-PINK1 (Abcam, ab23707, 1:2000), rabbit polyclonal anti-Cleaved Caspase-3 (CST, #9661, 1:2000), rabbit polyclonal anti-FLAG (CST, #2368, 1:2000 for IP samples), mouse monoclonal anti-FLAG (Sigma-Aldrich, F1804, 1:2000 for other samples), mouse monoclonal anti-α-tubulin (Sigma-Aldrich, T5168, 1:25,000), mouse monoclonal anti-β-actin (MBL, M177-3, 1:25,000), HRP-conjugated goat anti-rabbit IgG (Jackson ImmunoResearch, 111-035-003, 1:2000), HRP-conjugated goat anti-rat IgG (Jackson ImmunoResearch, 112-035-003, 1:2000) and HRP-conjugated goat anti-mouse IgG (Jackson ImmunoResearch, 115-035-003, 1:2000). The following antibody was used for immunohistochemistry at the indicated dilution: mouse monoclonal anti-PCNA (Santa Cruz Biotechnology, sc-56, 1:3000). The following antibody was used for immunocytochemistry at the indicated dilution: rabbit polyclonal anti-Tomm20 (Abcam, ab78547, 1:500), Alexa Fluor 488 goat anti-rabbit IgG (Abcam, ab150085, 1:1000). Bafilomycin A1 was purchased from Cayman Chemical.

**Animals.** C57BL/6J mice were obtained from the Nihon-CLEA. *Rubicon*-floxed mice[29] were previously generated in our laboratory. *Adipoq-Cre* mice[31] and *Atg5*-floxed mice[10] were obtained from Dr. Evan Rosen (Beth Israel Deaconess Medical Center) and Dr. Noboru Mizushima (The University of Tokyo), respectively. *Adipoq-Cre* mice were crossed with *Rubicon*-floxed mice or *Atg5*-floxed mice to generate mice harbouring homozygous deletion of *Rubicon* or *Atg5* specifically in adipocytes, respectively. *Rubicon*-floxed mice were crossed with *Atg5*ad−/− mice to generate mice with homozygous deletion of both *Rubicon* and *Atg5* specifically in adipocytes. All mice used in this study were not littermates because we used

*Adipoq-Cre* mice as controls to exclude the possibility that the phenotypes of the knockout mice arise from *Cre*-recombinase-mediated cellular dysfunction[69]. All kinds of transgenic mice were initially backcrossed into the C57BL/6J wild-type strain at least six times, and each mouse was generated from the same mouse colony maintained in the C57BL/6J background. The controls were maintained in the same mouse colony and same space. The control colony was used for the generation of the knockout strains. Just after weaning, age-matched mice were randomly co-housed independently from genotype, for the aging and HFD studies. No fighting was observed in the mice. The following primer sets were used for genotyping by PCR: 5′-ACAACGACAATCACACAGAC-3′ and 5′-TGAC-GAGGGGTAATGGATAG-3′ for *Rubicon* WT and floxed allele; 5′-ACAACGAC AATCACACAGAC-3′ and 5′-AATCCTTCGCCCCTTTTACC-3′ for *Rubicon* deleted allele; 5′-GAATATGAAGGCACACCCCTGAAATG-3′, 5′-GACAGGTCG GTCTTGACAAAAAGAAC-3′, and 5′-GTACTGCATAATGGTTTAACTCTT GC-3′ for *Atg5* WT and floxed allele; 5′-GCTCTTAGTCCCAGAACCTAAACC-3′ and 5′-GTACTGCATAATGGTTTAACTCTTGC-3′ for *Atg5*-deleted allele; 5′-GCATTACCGGTCGATGCAACGAGTGATGAG-3′ and 5′-GAGTGAAC-GAACCTGGTCGAAATCAGTGCG-3′ for *Cre*. Unless otherwise specified, all mice used for the experiments were 21-week-old males. These mice were maintained on an NCD in 12-h light/12-h dark cycles. The ambient temperature and humidity was 23 ± 1.5 °C and 45 ± 15%, respectively. Food and water were provided ad libitum. Sequence information about primers used for genotyping is available upon request. Five-week-old mice were fed with HFD for 16 weeks. Pioglitazone (Tokyo Chemical Industry) was administered at 0.02% (w/w) for 3 weeks from 21-week-old by admixing it with food. Experimental procedures using mice were approved by the Institutional Committee of Osaka University.

**Cell culture and plasmid transfection.** 3T3-L1 cells were purchased from National institute of Biomedical Innovation, JCRB Cell Bank. HEK293T cells were the same as one previously used in our laboratory[70]. Both cells were cultured in Dulbecco's modified Eagle's medium (Sigma-Aldrich, DMEM D6429) containing 10% foetal bovine serum (FBS) and 1% penicillin–streptomycin (Sigma-Aldrich, P4333) at 37 °C with 5% $CO_2$. Two days post confluence (defined as day 0), 3T3-L1 cells were treated for 48 h with adipogenic cocktail containing 0.5 mM 3-isobutyl-1-methylxanthine (Nacalai Tesque), 1 μM dexamethasone (Sigma-Aldrich), 1 μM insulin (Nacalai Tesque) and 10 μM pioglitazone (Wako) to induce adipogenesis. After that, the medium was replaced with DMEM with 10% FBS. Plasmid transfection was carried out using Opti-MEM (Gibco) and Lipofectamine 2000 (Invitrogen), according to the manufacturer's protocol. Pioglitazone (Wako) was used for activation of PPARγ. Valinomycin (Signa-Aldrich) was used for induction of mitophagy. EBSS (Sigma-Aldrich, E2888) was used for starvation treatment. Staining of ORO (Wako), TMRM (Tokyo Chemical Industry) and Hoechst33342 (Signa-Aldrich) was performed according to the standard protocol. All the cell lines were routinely tested for mycoplasma infection and confirmed as negative for mycoplasma contamination. Images were acquired with an IX83 (Olympus) or a BZ-X700 (Keyence). The ORO-stained area and the TMRM intensity were measured using a BZ-X700.

**Plasmids.** pmCherry-Parkin was purchased from Addgene. pENTR1A and pcDNA3.1 were purchased from Invitrogen. DNA fragments containing attR1, CmR, ccdB and attR2 were amplified from pAd/CMV/V5-DEST (Invitrogen) and subcloned into pcDNA3.1 (pcDNA3.1-DEST2). pcDNA3.1-DEST2-Nt-3xFLAG was generated from pcDNA3.1-DEST2 using In-Fusion (Takara). Full-length mouse SRC-1 and TIF2 sequences were amplified from mouse liver cDNA and cloned into pENTR1A (pENTR1A-WT-SRC-1 and pENTR1A-WT-TIF2). Mutant SRC-1 and TIF2 vectors were generated from wild-type vectors using In-Fusion (pENTR1A-W288A-SRC-1 and pENTR1A-W296A-TIF2). The sequences inserted into the pENTR1A plasmids were transferred into pcDNA3.1-DEST2-Nt-3xFLAG using Gateway LR Clonase (Invitrogen).

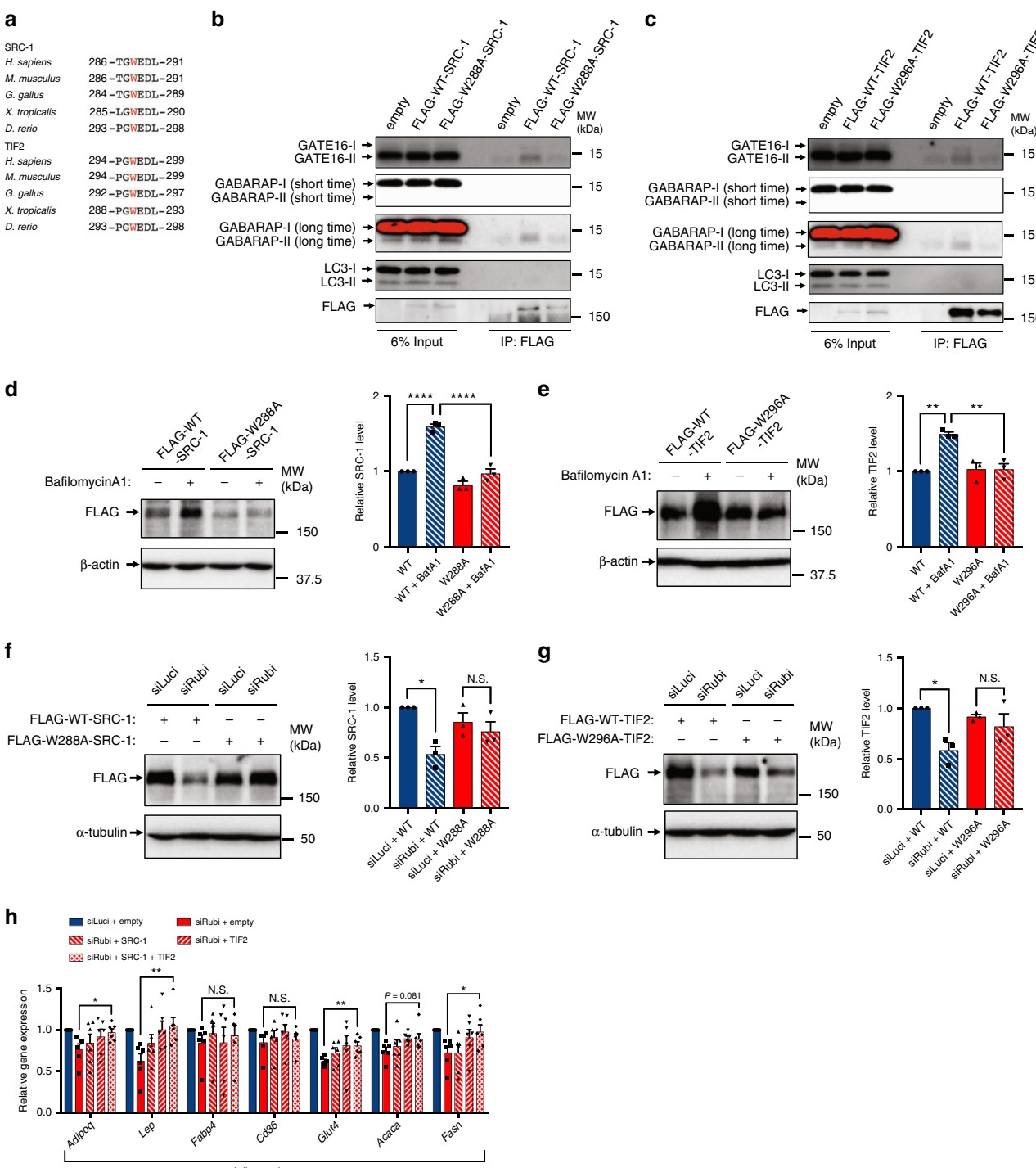

**Fig. 7 GABARAP family proteins are co-immunoprecipitated with SRC-1 and TIF2 in an LIR/GIM-dependent manner. a** Alignment of LIR/GIM of SRC-1 and TIF2 homologues in various species. **b, c** Immunoprecipitation assay. HEK293T cells were transfected with the indicated plasmids for 48 h, and then the cells were lysed and immunoprecipitated with anti-FLAG antibody. Precipitates were subjected to immunoblotting with the indicated antibodies. $n = 3$ independent experiments with similar results. **d, e** Immunoblotting to detect FLAG in the 3T3-L1 cells overexpressing the indicated plasmids. Twenty-four hours after transfection, the cells were treated with or without 125 nM Baf A1 treatment for 8 h. $n = 3$ independent experiments. **f, g** Immunoblotting to detect FLAG in the 3T3-L1 cells. Knockdown was performed for 48 h starting on day 8 following induction of differentiation. The indicated plasmids were transfected for 24 h starting on day 9. Cells were harvested at day 10. $n = 3$ independent experiments. **h** Relative mRNA expression of adipogenic genes in 3T3-L1 cells. Knockdown was performed for 48 h starting on day 8 following induction of differentiation. FLAG-W288A-SRC-1 and/or FLAG-W296A TIF2 plasmids were transfected for 24 h starting on day 9. FLAG-only plasmid was used as empty vector. Cells were harvested at day 10. $n = 6$ independent experiments. Quantification data are shown in the graphs at the right of each blot. Error bars indicate means ± SEM. Data were analysed by two-tailed Student's $t$ test (**h**), one-way ANOVA followed by Tukey's test (**d–g**). $P$ value from left to right: <0.0001, <0.0001 (**d**), 0.0013, 0.0019 (**e**), 0.0106, 0.8148 (**f**), 0.0170, 0.7972 (**g**), 0.0268, 0.0082, 0.5758, 0.6119, 0.0040, 0.0810, 0.0462 (**h**). *$P < 0.05$; **$P < 0.01$; ***$P < 0.001$. N.S. not significant.

**RNA interference**. siRNA duplex oligomers were purchased from Sigma-Aldrich. The design is as follows: 5′-UCGAAGUAUUCCGCGUACGdTdT-3′ (sense), 5′-CGUACGCGGAAUACUUCGAdTdT-3′ (antisense) for *Luciferase*; 5′-GAGCU-GAUGAAGUGCAACAUGAUGAGC-3′ (sense), 5′-UCAUCAUGUUGCACUU-CAUCAGCUCAA-3′ (antisense) for *Rubicon*. A total of 50 nM siRNA was introduced into cells using Opti-MEM (Gibco) and Lipofectamine RNAiMAX (Invitrogen), according to the manufacturer's instructions. The expression levels were assessed after 48 h by immunoblotting or qRT-PCR.

**Immunocytochemistry**. Cells were fixed with 4% PFA in PBS for 15 min at 37 °C to maintain mitochondrial morphology, and then were permeabilised with 50 µg/ml digitonin in PBS for 10 min at room temperature. Cells were washed once with PBS, and blocked with PBS containing 0.2% gelatin for 30 min at room temperature. Cells were incubated with the primary antibody in PBS containing 0.2% gelatin for 60 min at room temperature. After washed twice with PBS, cells were incubated with the secondary antibody and DAPI (Nacalai Tesque) in PBS containing 0.2% gelatin for 60 min. After washed twice with PBS, coverslips were mounted onto slides.

**Histological analyses**. Tissues were fixed in 4% paraformaldehyde overnight, and then stored in 70% ethanol until processing. Tissues were paraffinised and sectioned at 5 µm by microtome (Leica). The slides were stained with H&E according to a standard protocol. Immunohistochemical staining was performed on paraffin-embedded sections. After antigen retrieval by microwaving in unmasking buffer (10 mM sodium citrate, 0.05% Tween 20, pH 6.0) for 15 min, the sections were blocked with 2.5% Normal Horse Serum (Vector Laboratories) for 30 min at room temperature. The blocked sections were incubated with the primary antibody for 60 min at room temperature, followed by incubation for 60 min at room temperature with the secondary antibody, horse anti-mouse ImmPRESS (Vector Laboratories, MP-7402). Sections were counterstained with haematoxylin. DAB staining was performed with the DAB Peroxidase Substrate Kit, ImmPACT (Vector Laboratories). TUNEL staining was performed with In situ Apoptosis Detection Kit (Takara). DNase I-treated samples were used as a positive control. Images were acquired on a BX63 (Olympus) or a BZ-X700 (Keyence). Adipocyte size was measured on a BZ-X700.

**Metabolic studies**. Body weight and food intake were measured every week. Whole-body $O_2$ consumption and $CO_2$ production were monitored using an $O_2$/$CO_2$ metabolism measuring system for small animals (MK-5000RQ, Muromachi Kikai). In GTT experiments, 17-week-old mice were injected intraperitoneally with glucose (1 g per kg body weight) after a 4-h fast. In ITT experiments, 19-week-old mice were injected intraperitoneally with insulin (0.75 U per kg body weight) after a 4-h fast. Blood glucose levels were measured at the indicated time points using Glutest Neo alpha (Sanwa Kagaku). Unless otherwise specified, blood samples were collected under 4-h-fasted conditions. The following kits were used to determine metabolic parameters: Triglyceride Determination Kit (Wako), Total Cholesterol Kit (Wako), Free Fatty Acid Determination Kit (Wako), Insulin ELISA kit (Morinaga), Leptin ELISA kit (Morinaga) and Adiponectin ELISA kit (R&D Systems).

**Liver TG content**. Liver samples (50 mg) were homogenised in 1 ml Folch solution (2:1 v/v chloroform/methanol) using a Precellys Evolution tissue homogeniser (Bertin). Homogenates were added with 200 µl 0.9% NaCl solution. The lower phase was collected, and TG content determined using the Triglyceride Determination Kit (Wako).

**Cellular TG content**. Cells were harvested in 200 µl lysis buffer (25 mM Tris-HCl, pH 7.5, 1 mM EDTA, 1% Triton X-100). An aliquot of the lysate was mixed with the same amount of Folch solution (2:1 v/v chloroform/methanol). The lower phase was collected, and TG content determined, using a Triglyceride Determination Kit (Wako). Another aliquot of the lysate was used for protein determination by BCA assay (Nacalai Tesque).

**Lipolysis assay**. eWAT, iWAT and iBAT fat pads were isolated, weighed (about 50 mg for WAT, about 20 mg for BAT), cut into small pieces, and incubated in 600 µl of the lipolysis medium (DMEM (GIBCO 21063029), 2% BSA) with or without 10 µM isoproterenol (Sigma-Aldrich) for 3 h at 37 °C. After that, the medium was collected, and glycerol or NEFA concentration was measured using a free glycerol reagent (Sigma-Aldrich) or a Free Fatty Acid Determination Kit (Wako), according to the manufacturer's instructions, respectively. The content of Glycerol or NEFA was normalised to the initial tissue weight.

**RNA isolation and quantitative PCR analyses**. Mouse tissues were harvested in QIAzol (Qiagen) using a Precellys Evolution tissue homogeniser (Bertin). Total RNA was extracted using RNeasy Plus Mini kit (Qiagen). cDNA was generated using iScript (Bio-Rad). qRT-PCR was performed with *Power* SYBR Green (Applied Biosystems) on a QuantStudio 7 Flex Real-time PCR System (Applied Biosystems). Four technical replicates were performed for each reaction. The data were excluded from qPCR if >30 cycles were performed without amplification

signal. *36b4* was used as an internal control. Sequences of qRT-PCR primers are shown in Supplementary Table 1.

**Immunoblotting**. Mouse tissues were harvested in RIPA buffer [50 mM Tris-HCl pH 8.0, 150 mM NaCl, 1% w/v Triton X-100, 0.1% SDS, 0.5% sodium deoxycholate, protease inhibitor cocktail (Roche)] using a Precellys Evolution tissue homogeniser, and cells were lysed in the same buffer. After centrifugation, the supernatants were subjected to protein quantification by BCA assay (Nacalai Tesque). Lysates were mixed with 5× SDS sample buffer and boiled for 5 min. Protein lysates were separated by 7 or 13% SDS–PAGE and transferred to PVDF membranes. Membranes were stained with Ponceau S, which were then blocked with 1% skim milk TBS-T and incubated with specific primary antibodies. Immunoreactive bands were detected using HRP-conjugated secondary antibodies, visualised with Luminata Forte (Merck Millipore) or ImmunoStar LD (Wako), and imaged using a ChemiDoc Touch (Bio-Rad). α-tubulin or β-actin was used as a loading control. The band intensity of each protein was normalised against the loading control for quantification. Band intensity was quantified using the ImageJ software (NIH).

**Immunoprecipitation assay**. Cells were lysed in lysis buffer [20 mM HEPES-NaOH, pH 7.5, 150 mM NaCl, 1.0% digitonin, protease inhibitor cocktail (Roche)]. After centrifugation, the resultant supernatants were quantified by BCA assay. The samples were incubated overnight at 4 °C with anti-FLAG M2 affinity gel (Sigma-Aldrich), or Protein G Sepharose 4 Fast Flow (GE Healthcare) with rabbit polyclonal anti-p62 (MBL, PM045). After repeated washing, the bound proteins were eluted with SDS sample buffer, and then analysed by SDS–PAGE and immunoblotting.

**Autophagy flux assay**. In vitro, cells were incubated for 2 h in normal medium with or without 125 nM Bafilomycin A1 at 37 °C with 5% $CO_2$. Cells were lysed and immunoblotted for LC3 or p62. Ex vivo, freshly collected tissue explants were incubated for 2 h in high-glucose DMEM with or without 20 mM ammonium chloride and 200 µM leupeptin at 37 °C with 5% $CO_2$[21]. Explants were processed and immunoblotted for LC3. Autophagic flux was calculated by subtracting the densitometry values of LC3-II or p62 in samples not treated with lysosomal inhibitor from the corresponding values in lysosomal inhibitor-treated samples.

**Flow cytometry**. eWAT was minced in 10 ml of the FACS buffer (DMEM (GIBCO 21063029) with 10% FBS and antibiotics). After centrifugation, the supernatant was added with 500 µl of Collagenase (10 mg/ml) and 500 µl of DNase (2 mg/ml), and was incubated at 37 °C for 30 min under constant shaking. The cell suspension was filtered through a 70-µm cell strainer and then centrifuged to separate the SVF pellet from the floating mature adipocytes fraction. The SVF cells were suspended in the FACS buffer and incubated with anti-mouse CD16/CD32 (Tonbo, 70-0161, 1:100) for 20 min. Then, the cells were stained for 30 min with PerCP anti-CD45 (BioLegend, 103129, 1:100), FITC anti-CD11b (BioLegend, 101205, 1:100) and PE anti-F4/80 (eBioscience, 12-4801-82, 1:100). The SVF cells were washed three times and resuspended with the FACS buffer, and analysed with SH800S (SONY). Cellular debris was excluded from analysis. Single-stained and unstained controls were used for compensation and gating. The same gate was applied to all samples.

**Statistical analyses**. All results are presented as means ± S.E.M. Statistical analysis were performed with two-tailed Student's *t* test, one-way ANOVA followed by Tukey's test or Dunnett's test, or two-way repeated-measures ANOVA followed by Fisher's LSD test or Tukey's test using Excel for Mac (Microsoft) and GraphPad Prism7 (GraphPad Software).

**Reporting summary**. Further information on research design is available in the Nature Research Reporting Summary linked to this article.

## Data availability

We used iLIR database (http://repeat.biol.ucy.ac.cy/iLIR) for investigating LIR/GIM-like motifs. Source data are provided with this paper. Uncropped scans of the blots are included in the Source data file. All data are also available from the corresponding author upon reasonable request.

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

## Acknowledgements

We thank Dr. Evan Rosen (Beth Israel Deaconess Medical Center) for *Adipoq-Cre* mice, and Dr. Noboru Mizushima (The University of Tokyo) for *Atg5*-floxed mice. T.Ya. was supported by the Takeda Science Foundation. T.Yo. was supported by MEXT/JSPS KAKENHI, the Takeda Science Foundation, the HFSP grant and the A3 Foresight Program. This research was supported by Japan Agency for Medical Research and Development, AMED under grant numbers JP18gm5010001 and JP17gm0610005.

## Author contributions

T.Ya., T.K., A.F., I.S., and T.Yo. designed the study. T.Ya. performed mouse experiments with the help of T.K., A.F., M.F., Y.E., G.Y., and K.T. H.T. and K.Y. generated 25-month-old wild-type mice. T.Ya. conducted cell culture experiments. S.S., S.N., M.H. and A.K. contributed valuable discussions. T.Ya., T.K., A.F., I.S. and T.Yo. wrote the paper.

## Competing interests

I.S. and T.Yo. have applied for the patent related to this work. T.Yo. is the founder for AutoPhagyGO. A.F. belongs to the department endowed by Takeda Pharmaceutical Company, Rohto Pharmaceutical Co., Ltd, Sanwa Kagaku Kenkyusho Co., Ltd, Fuji Oil Holdings Inc. and Kobayashi Pharmaceutical Co., Ltd. All other authors declare no competing interests.
