## [Peer Review File · Nature Communications]

Peer Review File - Reviewers' comments first round:

Reviewer #1 (Remarks to the Author):

In this manuscript, Yamamuro et al explore the unknown mechanisms underlying adipocyte dysfunction in aged organisms. The authors explored the role of autophagy, which is required for adipocyte differentiation, in adipocyte function. Interestingly, aged adipocytes display increased autophagic activity, which is counter to reports from other tissues. Thus the authors explored the role of Rubicon, a negative regulator of autophagy, in adipocyte function.

Major

Rubicon is required for LC3-associated phagocytosis (LAP), lipophagy is a critical regulator of adipocyte homeostasis, and mitophagy is a critical regulator of browning of WAT. Yet the authors do not even mention LAP and do not explore other types of non-canonical autophagy, specifically mitophagy or lipophagy. How do the authors know that this is canonical autophagy? More experiments are needed to elucidate the correct autophagic pathway.

At what age does Rubicon start to decline? 3 and 25 month are two extremes and timepoints between the 2 should be included.

Autophagy is known to decrease with age, and is associated with age-related complications. However, these authors demonstrate that, in adipose tissue, Rubicon is decreased with age, and subsequently, autophagy is increased. Can the authors speculate why adipose tissue acts counterintuitively to other tissues? Decreases in autophagic machinery in other aged tissues is associated with increased damage or decreased function, yet here increased autophagy is associated with aged adipocytes and forcing this phenomenon with a Rubicon-Ad-Cre model seems to result in metabolic disorder? Do these mice also exhibit any cardiovascular defects?

Is the differentiation of adipocyte normal in younger Rubicon-Ad-Cre mice?

Are Rubicon-Ad-Cre adipocytes undergoing cell death, perhaps autophagic cell death?

It would be helpful to quantify the number and size of both WAT and BAT. Is this merely a hypertrophic effect?

Could the authors quantify and expand upon their hypothesis that Rubicon-Ad-Cremice have abnormal LD format?

FACS should be performed to characterize inflammatory cell infiltration in WAT as well as BAT

Rubicon-Ad-Cre mice should be treated with PPAR ligand or IL-10 overexpression mouse.

The question remains – how is increased autophagy interfering with adipocyte function? Is it shuttling LD out of adipocytes? Does Rubicon control transcription of these adipogenic genes directly? The authors fail to bring the story to a close and explore the bigger picture, which leaves this reviewer unsatisfied.

Minor

This manuscript requires serious proofreading for grammar, clarity, and syntax.

Authors should use one consistent reference style.

What age are tissues in Figure 1d?

What about BAT levels of Rubicon, autophagy players? Only Rubicon and p62 are explored, and other autophagy players should also be assessed.

Reviewer #2 (Remarks to the Author):

In this work, Yamamuro et al. attempt to draw a functional and causal relationship between Rubicon, an inhibitor of autophagy, and adipose biology. Although the adipose-specific Rubicon KO mice could eventually prove to be an interesting model to study the role of autophagy in adipose tissue function, the studies herein are far too preliminary to provide any new light on this link. There are concerns related to missing controls to show how autophagy behaves in adipose tissue lacking Rubicon. The metabolic consequences of loss of Rubicon are quite poorly characterized as detailed below. The reason to assess differentiation in adipose-specific KO mice by using a mouse-line where Cre is expressed post-differentiation remains unclear. Finally, it remains unexplained why hyperactivating autophagy (by knocking out Rubicon) and knocking out Atg7 (as shown previous) will each impact adipocyte differentiation. Extensive studies at multiple time-points will be needed to dissect the role of Rubicon in adipose tissue function.

1. Figure 1a. qPCR control for p62 to confirm that decreased p62 protein levels in aged mice is not due to decreased gene expression. The authors must also include blots for p62 and LC3 in presence and absence of lysosome inhibitors in vivo (i.p.) to confirm that autophagy is indeed hyperactivated in aged adipose tissue.
2. In supplementary Fig. S1, the authors should demonstrate that autophagy/LC3-II flux is induced in the adipose-specific Rubicon KO mice. The reviewer understands that loss of Rubicon will lead to the upregulation of autophagy, and appreciates that Rubicon/Atg5 double KO mice show blocked autophagy; however, the degree to which autophagy is induced in Rubicon KO mice in distinct adipose depots (eWAT, iWAT) will be interesting for the field to know and should be characterized.
3. White adipocyte sizes between the different groups should be (e.g., Fig 1) quantified.
4. What is the reason for the elevation of serum cholesterol in the adipose-specific Rubicon KO mice?
5. Metabolic characterization of Rubicon KO mice is inadequate in terms of the changes in TG metabolism shown. Adipose TG breakdown (lipolysis) increases during fasting and under these conditions, significant lipid is mobilized from adipocytes to the liver. Given that this phenomenon is replicated in the adipose Rubicon KO mice under basal state, it is possible that lipophagy/lipolysis (lysosomal and neutral) and fat mobilization are hyperactivated in adipose tissue of Rubicon KO mice. This should be characterized to provide a more comprehensive view of this model. Are lipolytic pathways induced in adipose tissue? Whole body energy expenditure rates (as shown in supplemental data) may not be sensitive enough to pick up depot-specific increases in lipolysis/fat oxidation. It would be particularly useful to carry out these experiments in both and fasted Rubicon and Rubicon/Atg5 dual KO mice.
6. What is the effect of a high fat diet feeding on adipose-specific Rubicon KO mice?
7. Does loss of Rubicon lead to adipocyte death or defects in adipocyte proliferation? Some of these assessments will greatly help understand this mouse model of hyperactive autophagy.
8. Concerns related to differentiation experiments: In vitro assessments shown in Fig 3 are also rather preliminary, and it would be more useful to look at changes in autophagy flux and Rubicon levels on day 0, 2, 4, 6, 8, 10, 12 and 16 after initiation of differentiation – revealing details as to when Rubicon expression begins in course of differentiation. On day 10, adipocytes already show significant increase in markers of terminal differentiation, e.g., FAS and ap2, and it is possible that Rubicon is only expressed after the adipocyte is differentiated. This was perhaps the rationale for the authors to choose the adiponectin-Cre line to delete Rubicon in mature adipocytes. As a consequence, data showing impaired differentiation in mature adipocytes knocked out for Rubicon (using the adiponectin-Cre) is rather confusing and difficult to interpret. Furthermore, there are studies that have shown that autophagy is required for muscle and adipocyte differentiation, while this work would suggest that loss of Rubicon, and as a consequence, hyperactivation of autophagy impairs differentiation. How do the authors explain these findings?
9. The rationale to look at PPAR agonists in Rubicon KO mice remains unclear.
10. Blots in some of the figure panels should have been run in the same gel.

Point-by-point response

Reviewer #1:

General comments

In this manuscript, Yamamuro et al explore the unknown mechanisms underlying adipocyte dysfunction in aged organisms. The authors explored the role of autophagy, which is required for adipocyte differentiation, in adipocyte function. Interestingly, aged adipocytes display increased autophagic activity, which is counter to reports from other tissues. Thus the authors explored the role of Rubicon, a negative regulator of autophagy, in adipocyte function.

Thank you for your comments. Because the comments were insightful and constructive, we could greatly improve our study with the additional experiments. Following is the detailed response to the comments.

Major

Comment-1

Rubicon is required for LC3-associated phagocytosis (LAP), lipophagy is a critical regulator of adipocyte homeostasis, and mitophagy is a critical regulator of browning of WAT. Yet the authors do not even mention LAP and do not explore other types of non-canonical autophagy, specifically mitophagy or lipophagy. How do the authors know that this is canonical autophagy? More experiments are needed to elucidate the correct autophagic pathway.

Reply-1

Thank you for the comment. We agree with the idea that it is critical to dissect what kind of autophagic process is a major contributor to the phenotypes of *Rubicon*^{ad-/-} mice. Thus, we carried out the experiments to determine whether mitophagy or lipophagy in adipocytes occurs in normal mice. As shown below, we performed *ex vivo* culture of adipose tissue with a lysosomal inhibitor to monitor the autophagic flux of the potential substrates of these pathways. While an autophagic substrate LC3-II showed a significant accumulation in the presence of the inhibitor, we did not observe any detectable increase in Perilipin1 or Perilipin2 (Supplementary Fig. 6h), which are potential substrates of lipophagy in adipose tissue, suggesting that basal lipophagy does not occur at a detectable level (To avoid any confusion, we would like to mention that Perilipin1 and Perilipin2 were downregulated at the transcriptional level in *Rubicon*^{ad-/-} mice (Supplementary Fig. 6a, b)). Complex III subunit 1, a substrate of mitophagy, also did not increase in the presence of the inhibitor (Supplementary Fig. 6h). These results indicate that mitophagy or lipophagy in adipocytes could play only a limited role in normal mice, whereas previous reports showed that two selective autophagy pathways play a critical role in adipocytes of cold-exposed mice (N Martinez-Lopez *et al.*, *Cell Metab.* 2016, S Altshuler-Keylin *et al.*, *Cell Metab.* 2016). They do not exclude the possibility that those mitophagy or lipophagy is a minor contributor to the phenotypes of *Rubicon*^{ad-/-} mice. Strikingly, we found that canonical autophagy is a major contributor to the phenotypes. As mentioned below, we found that two coactivators of PPAR γ , SRC-1 and TIF2, were directly degraded by autophagy through their

binding to GABARAP family proteins, and were significantly downregulated in adipocytes of *Rubicon^{ad-/-}* mice (Fig. 6, 7). Taken together, our findings indicate that an excessive increase in canonical autophagy is a major factor leading to metabolic disorders in *Rubicon^{ad-/-}* mice. We are sorry for that it was not clearly mentioned in the previous manuscript that a potential decrease in LAP caused by loss of Rubicon is not the contributor. This is because the phenotypes were cancelled by additional knockout of *Atg5*, which is an essential factor for LAP as well as Rubicon. We mentioned it in the revised manuscript (Line number: 115–118).

Fig. S6a

Fig. S6b

Fig. S6h

Comment-2

At what age does Rubicon start to decline? 3 and 25 months are two extremes and timepoints between the 2 should be included.

Reply-2

To determine the starting point of the decline in Rubicon, we carried out an additional experiment, and found that the decline was seen at the age of 12 months. As shown also in our previous manuscript, Rubicon was downregulated in adipocytes of 25-month-old mice (Fig. 1a), and this reduction was evident even in 12- and 18-months-old mice. (Fig. 3a). Thus, we could speculate that a decline in Rubicon precedes the metabolic disorders in aged individuals. We appreciate this valuable comment.

Fig. 1a

Fig. 3a

Comment-3

Autophagy is known to decrease with age, and is associated with age-related complications. However, these authors demonstrate that, in adipose tissue, Rubicon is decreased with age, and subsequently, autophagy is increased. Can the authors speculate why adipose tissue acts counterintuitively to other tissues? Decreases in autophagic machinery in other aged tissues is associated with increased damage or decreased function, yet here increased autophagy is associated with aged adipocytes and forcing this phenomenon with a Rubicon-Ad-Cre model seems to result in metabolic disorder? Do these mice also exhibit any cardiovascular defects?

Reply-3

We appreciate your insightful suggestions. We realised that it is worth emphasising that adipocytes are unique in terms of autophagy during aging process. We are sorry for that we still could not fully answer the question of why aged adipocytes exhibit higher autophagic flux while other tissues show a decline in autophagic activity during aging process. It is clear that an age-dependent decline in Rubicon leads to upregulation of autophagy, but the mechanism of which remains unclear. Because Rubicon was downregulated at transcriptional level (Fig. 3b), we could speculate that an adipocyte-specific transcriptional regulation contributes to the loss of Rubicon. We think this discussion should be only for the reviewing process because it is highly speculative. We will address the issue in the future.

Answering the question of why upregulation of autophagy has a beneficial effect on other tissues but causes a deleterious effect in adipocytes, we found that SRC-1 and TIF2, which are coactivators of PPAR γ , in adipocytes were degraded by

autophagy (Fig. 6a, b, d). They were downregulated in *Rubicon*^{ad-/-} mice and aged mice (Fig. 6c, e, f), suggesting that excess autophagic degradation of SRC-1 and TIF2 is a major cause of adipocyte dysfunction in *Rubicon*^{ad-/-} mice. This mechanism seems to be a disadvantage for normal individuals. But it could be explained by the fact that SRC-1 and TIF2 in adipose tissue were downregulated in fasting mice (Fig. 6g). Of course, adipocytes should not store the energy during fasting. Thus, we speculate that autophagic degradation of SRC-1 and TIF2 actively leads to a decline in adipocyte function during fasting, but in turn it leads to harmful consequences during aging process. We consider this trade-off is the underlying mechanism of how *Rubicon*^{ad-/-} mice or aged mice show the metabolic disorders. Notably, we did not find any cardiovascular defect in *Rubicon*^{ad-/-} mice, and at least the heart weight was not changed (Review only Fig. 1).

Fig. 6a

Fig. 6b

Fig. 6c

Fig. 6e

Fig. 6d

Fig. 6g

Fig. 6f

Fig. 3c

Review only Fig. 1 (n = 10)

Comment-4

Is the differentiation of adipocyte normal in younger Rubicon-Ad-Cre mice?

Reply-4

We observed the decline in adipocyte function even in younger *Rubicon*^{ad-/-} mice. As well as 5-month-old mice, adipogenic genes in 3-month-old *Rubicon*^{ad-/-} mice were also decreased (Supplementary Fig. 6i). Consistent with this, *Rubicon*^{ad-/-} mice weighed less than control mice at 3-month-old (Fig. 1b), also suggesting that Rubicon is essential for adipogenesis in younger mice.

Fig. S6i

Fig 1b

Comment-5

Are Rubicon-Ad-Cre adipocytes undergoing cell death, perhaps autophagic cell death?

Reply-5

We appreciate the valuable comment. Because accumulating evidence have shown that autophagy can promote cell death including apoptosis or necroptosis (J Doherty and EH Baehrecke, *Nat Cell Biol* 2018), we agree with the idea that adipocyte dysfunction by *Rubicon* deletion could stem from potential cell death that is promoted by upregulation of autophagy. Thus, we conducted several analyses, and found that adipocytes in *Rubicon*^{ad-/-} mice would not undergo cell death. TUNEL-positive cells were detected in a DNase-treated sample, used as a positive control, but not in untreated samples from control mice or *Rubicon*^{ad-/-} mice (Supplementary Fig. 7c). Consistent with this, *Rubicon* knockout in adipocytes did not increase Cleaved Caspase-3 (Supplementary Fig. 7b). CLSs, dead adipocytes surrounded by macrophages (S Cinti *et al.*, *J. Lipid Res.* 2005), were not observed in the eWAT in either control or *Rubicon*^{ad-/-} mice, while they were detected in HFD-fed mice (Supplementary Fig. 7d), indicating that cell death is not induced in adipocytes of *Rubicon*^{ad-/-} mice. Based on these findings, we conclude that, even if loss of Rubicon could potentially increase cell death at an undetectable level, cell death is not a major contributor to adipocyte dysfunction in *Rubicon*^{ad-/-} mice.

Fig. S7b

Fig. S7d

Fig. S7c

Comment-6

It would be helpful to quantify the number and size of both WAT and BAT. Is this merely a hypertrophic effect?

Reply-6

We acknowledge the suggestion. It is helpful to measure the number and size of adipose tissues. Basically, we consider that *Rubicon* knockout has an atrophic effect on adipocytes. We quantified the size of white adipocytes in NCD-fed mice, HFD-fed mice, and aged mice, and confirmed that *Rubicon* knockout decreases the adipocyte size in these mice (Fig. 1g, 3h, i and Supplementary Fig. 4d). The cell body size or LD size of brown adipocytes was also decreased in NCD-fed mice, HFD-fed mice, and aged mice (Fig. 1h and Supplementary Fig. 4e, 5d). Because we observed a significant reduction in the adipocyte size in *Rubicon*^{ad-/-} mice, we consider that it is not adequate to quantify the number of cells per area to interpret the cause of the atrophy. However, as mentioned before, we observed no increase in adipocyte death in *Rubicon*^{ad-/-} mice, supporting the idea that the reduction in adipose mass of *Rubicon*^{ad-/-} mice stems from a decrease in adipocyte size, but not in adipocyte number.

Fig. 1g

Fig. 3h

Fig. 3i

Fig. S4d

Fig. 1h

Fig. S4e

Fig. S5d

Comment-7

Could the authors quantify and expand upon their hypothesis that Rubicon-Ad-Cre mice have abnormal LD format?

Reply-7

Yes, we observed that *Rubicon^{ad-/-}* mice have small LDs; therefore, the fat atrophy in *Rubicon^{ad-/-}* mice stems from the abnormal LD. As mentioned above, *Rubicon^{ad-/-}* mice exhibited a reduction in LD size in eWAT and iBAT (Fig. 1f-h). We also found that Perilipin1 and Perilipin2, which participate in LD stabilization or lipolysis, were downregulated at a transcriptional level in *Rubicon^{ad-/-}* mice (Supplementary Fig. 6a, b), suggesting that the decreases in Perilipin1 and Perilipin2 could lead to the reduction in LD size. A previous study showed that Perilipin1 is regulated by PPAR γ (N Arimura *et al.*, *J. Biol. Chem.* 2004); thus, the reduction in Perilipin1 and Perilipin2 could arise from a decline in PPAR γ activity in *Rubicon^{ad-/-}* mice.

Fig. 1f

Fig. 1g

Fig. S6b

Fig. 1h

Fig. S6a

Comment-8

FACS should be performed to characterize inflammatory cell infiltration in WAT as well as BAT

Reply-8

Thank you for the insightful suggestion. We agree with the idea that adipose inflammation should be closely examined in *Rubicon^{ad-/-}* mice. Prior to a detailed analysis of inflammatory cell infiltration by FACS, we sought to determine whether systemic inflammation is induced in the mice. We found that the plasma IL-6 and TNF levels in *Rubicon^{ad-/-}*

mice were comparable to those in control mice (Supplementary Fig. 7e, f), suggesting that systemic inflammation is not induced in *Rubicon*^{ad-/-} mice. Moreover, we determined whether inflammatory cell infiltration occurs in WAT and BAT. To this end, we performed qRT-PCR assay, and found that the gene expression of pro-inflammatory cytokines or macrophage markers was not significantly elevated in eWAT or iBAT of *Rubicon*^{ad-/-} mice (Supplementary Fig. 7g-j). Histological analysis revealed that inflammatory cells were not seen in the eWAT or the iBAT in *Rubicon*^{ad-/-} mice (Fig. 1f, h), and that CLSs in the eWAT were detected in HFD-fed mice, but not in *Rubicon*^{ad-/-} mice (Supplementary Fig. 7d). These results suggest that inflammatory cell infiltration does not occur in adipose tissue of *Rubicon*^{ad-/-} mice, even if we did not perform FACS. Thus, we conclude that inflammation is not a major contributor to the phenotypes of *Rubicon*^{ad-/-} mice.

Fig. S7d

Fig. S7e

Fig. S7f

Fig. S7g

Fig. S7h

Fig. S7i

Fig. S7j

Fig. 1f

Fig. 1h

Comment-9

Rubicon-Ad-Cre mice should be treated with PPAR ligand or IL-10 overexpression mouse.

Reply-9

We treated *Rubicon*^{ad-/-} mice with a PPAR γ ligand TZD, and found that TZD treatment recovered the endocrine dysfunction in *Rubicon*^{ad-/-} mice (Fig. 5e, f). Importantly, TZD treatment also rescued the fat loss in the knockout mice (Fig. 1d, 5g). Thus, as well as *in vitro*, activation of PPAR γ by TZD reverses adipocyte dysfunction upon *Rubicon* knockout *in vivo*.

Comment-10

The question remains – how is increased autophagy interfering with adipocyte function? Is it shuttling LD out of adipocytes? Does Rubicon control transcription of these adipogenic genes directly? The authors fail to bring the story to a close and explore the bigger picture, which leaves this reviewer unsatisfied.

Reply-10

Thank you for your comment. We realised that this is the biggest issue in the previous manuscript. We paid a special attention to this matter, and determined the mechanism of how excess autophagy leads to adipocyte dysfunction. As mentioned above, we found that upregulation of autophagy interferes with adipocyte function via degradation of SRC-1 and TIF2. SRC-1 and TIF2 were degraded by autophagy, and were significantly downregulated in adipose tissue of *Rubicon*^{ad-/-} mice (Fig. 6b–d). This phenomenon was also seen in *Rubicon*-knockdown cells (Fig. 6a). In addition, we found potential LIR/GIMs in SRC-1 and TIF2 by using iLIR database (Fig. 7a), and confirmed that GABARAP family proteins were immunoprecipitated with FLAG-tagged SRC-1 or TIF2 in a LIR/GIM-dependent manner (Fig. 7b, c). Consistent with this, the LIR/GIM mutant SRC-1 and TIF2 failed to exhibit an increase in the presence of a lysosomal inhibitor (Fig. 7d, e), as well as a reduction upon *Rubicon* depletion (Fig. 7f, g), suggesting that autophagic degradation of SRC-1 and TIF2 by *Rubicon* depletion is mediated by the interaction with GABARAP family proteins. Importantly, both overexpression of the LIR/GIM mutant SRC-1 and TIF2 rescued the reduction in adipogenic gene expressions in

Rubicon-knockdown cells (Fig. 7h). Taken together, we propose that excess autophagy in adipocytes causes the reduction in SRC-1 and TIF2, leading to a decline in PPAR γ activity and adipocyte function in *Rubicon*^{ad-/-} mice.

Fig. 7a

SRC-1	
H. sapiens	286-TGWEDL-291
M. musculus	286-TGWEDL-291
G. gallus	284-TGWEDL-289
X. tropicalis	285-LGWEDL-290
D. rerio	293-PGWEDL-298
TIF2	
H. sapiens	294-PGWEDL-299
M. musculus	294-PGWEDL-299
G. gallus	292-PGWEDL-297
X. tropicalis	288-PGWEDL-293
D. rerio	293-PGWEDL-298

Fig. 7b

Fig. 7c

Fig. 7d

Fig. 7e

Fig. 7f

Fig. 7g

Fig. 7h

Minor

Comment-11

This manuscript requires serious proofreading for grammar, clarity, and syntax.

Reply-11

We have asked the proofreader, and our manuscript has been revised.

Comment-12

Authors should use one consistent reference style.

Reply-12

We have revised this point.

Comment-13

What age are tissues in Figure 1d?

Reply-13

We obtained the tissues in previous Figure 1d from 21-week-old mice, and found that *Rubicon* knockout in adipocytes reduced fat mass in 21-week-old mice (Fig. 1d). We added the information for age in Figure legends. We also found that adipose weight was significantly decreased in 12 and 18-month-old *Rubicon*^{ad-/-} mice (Fig. 3e, f, Supplementary Fig. 5a–c). These results indicate that loss of adipose *Rubicon* causes fat reduction during aging process.

Fig. 1d

Fig. 3e

Fig. 3f

Fig. S5a

Fig. S5b

Fig. S5c

Comment-14

What about BAT levels of Rubicon, autophagy players? Only Rubicon and p62 are explored, and other autophagy players should also be assessed.

Reply-14

We have checked Rubicon, p62, and other ATG proteins in WAT and BAT of *Rubicon^{ad-/-}* mice. We confirmed Rubicon decreased in the iBAT of *Rubicon^{ad-/-}* mice (Supplementary Fig.1e), and found that LC3-II and p62 in BAT were significantly downregulated as well as eWAT (Supplementary Fig.2d). Other ATG proteins in eWAT or iBAT were not significantly changed in *Rubicon^{ad-/-}* mice (Supplementary Fig.2a, b).

Fig. S1e

Fig. S2a

Fig. S2b

Fig. S2d

Reviewer #2 (Remarks to the Author):

General comments

In this work, Yamamuro et al. attempt to draw a functional and causal relationship between Rubicon, an inhibitor of autophagy, and adipose biology. Although the adipose-specific Rubicon KO mice could eventually prove to be an interesting model to study the role of autophagy in adipose tissue function, the studies herein are far too preliminary to provide any new light on this link. There are concerns related to missing controls to show how autophagy behaves in adipose tissue lacking Rubicon. The metabolic consequences of loss of Rubicon are quite poorly characterized as detailed below. The reason to assess differentiation in adipose-specific KO mice by using a mouse-line where Cre is expressed post-differentiation remains unclear. Finally, it remains unexplained why hyperactivating autophagy (by knocking out Rubicon) and knocking out Atg7 (as shown previous) will each impact adipocyte differentiation. Extensive studies at multiple time-points will be needed to dissect the role of Rubicon in adipose tissue function.

We appreciate your insightful comments. We believe that the reviewer raised essential questions that should be clearly addressed upon publication. We seriously considered the questions, and conducted the experiments to answer all of those issues. Following is the detailed response to the comments.

Comment

1. Figure 1a. qPCR control for p62 to confirm that decreased p62 protein levels in aged mice is not due to decreased gene expression. The authors must also include blots for p62 and LC3 in presence and absence of lysosome inhibitors in vivo (i.p.) to confirm that autophagy is indeed hyperactivated in aged adipose tissue.

Reply-1

We appreciate the valuable comment. We conducted the new experiments, and confirmed that aged mice exhibited upregulation of autophagy in adipose tissue. Systemic inhibition of autophagic activity by intraperitoneal injection of a lysosomal inhibitor could indirectly affect the autophagic flux in adipocytes via any change in other organ(s); therefore, we employed *ex vivo* autophagic flux assay to assess the correct autophagic activity in adipose tissue (N Martinez-Lopez *et al.*, *Cell Metab.* 2017). As a result, *ex vivo* autophagy flux assay using LC3-II revealed that autophagic activity in adipose tissue increased in aged mice (Fig. 3c). Any increase in p62 by a lysosomal inhibitor was not detected even in adipose tissue of wild-type mice (Review only Fig. 2). This could be due to that a degradation rate of p62 in adipocytes is intrinsically too low to be detected by the flux assay in which lysosomal activity is inhibited in limited duration. In addition, we found that Rubicon and p62 were significantly decreased in adipose tissue from 12- or 18-month old mice while the gene expression of p62 was not decreased in both mice (Fig. 3a, b), suggesting that the age-dependent reduction of p62 is not due to a decrease in the gene expression. Based on these findings, we conclude that autophagic activity in adipocytes increases with age.

Fig. 3a

Fig. 3b

Fig. 3c

Review only Fig. 2

Comment

2. In supplementary Fig. S1, the authors should demonstrate that autophagy/LC3-II flux is induced in the adipose-specific Rubicon KO mice. The reviewer understands that loss of Rubicon will lead to the upregulation of autophagy, and appreciates that Rubicon/Atg5 double KO mice show blocked autophagy; however, the degree to which autophagy is induced in Rubicon KO mice in distinct adipose depots (eWAT, iWAT) will be interesting for the field to know and should be characterized.

Reply-2

Thank you for the comment. We considered that autophagy in white adipocytes and brown adipocytes should be assessed, and confirmed that the autophagic activity in the eWAT and the iBAT increased in *Rubicon*^{ad-/-} mice. By using *ex vivo* autophagic flux assay, we found that autophagic activity was clearly increased in the eWAT and iBAT in *Rubicon*^{ad-/-} mice (Supplementary Fig. 2e, f). Consistent with this, as well as in eWAT, LC3-II and p62 in iBAT were significantly decreased in *Rubicon*^{ad-/-} mice (Supplementary Fig. 2d). These results indicate that autophagy is upregulated in both white and brown adipocytes in *Rubicon*^{ad-/-} mice.

Fig. S2d

Fig. S2e

Fig. S2f

Comment

3. White adipocyte sizes between the different groups should be (e.g., Fig 1) quantified.

Reply-3

We quantified white adipocyte size in NCD-fed mice, HFD-fed mice, and aged mice (Fig. 1g, 3h, i and Supplementary Fig. 4d), and confirmed that *Rubicon* knockout causes a reduction in adipocyte size in these mice.

Fig. 1g

Fig. 3h

Fig. 3i

Fig. S4d

Comment

4. What is the reason for the elevation of serum cholesterol in the adipose-specific Rubicon KO mice?

Reply-4

Thank you for the comment. We consider that both of a reduction in LPL and increases in ABCA1 and ABCG1 could be a reason for an elevation in plasma cholesterol in *Rubicon^{ad-/-}* mice. We found that the gene expression of LPL, a lipase for lipoprotein, was significantly decreased in the eWAT in *Rubicon^{ad-/-}* mice (Supplementary Fig. 6c). This reduction is most likely mediated by a decline in PPAR activity because PPAR γ positively regulates *Lpl* expression. As shown previously, knockout of *Lpl* causes an increase in both plasma triglyceride and cholesterol (PH Weinstock *et al.*, *J Clin Invest.* 1995); thus, the reduction in LPL could lead to the elevation of not only plasma triglyceride, but also plasma cholesterol in *Rubicon^{ad-/-}* mice. In addition, we found that the gene expression of *Abca1* and *Abcg1*, both of which participate in cholesterol efflux, was elevated in *Rubicon^{ad-/-}* mice (Supplementary Fig. 6c). A recent study showed that adipose-specific knockout of *Abca1* reduces plasma cholesterol (W de Haan *et al.*, *J. Lipid Res.* 2014). Therefore, the increment in ABCA1 and ABCG1 could contribute to the elevation of plasma cholesterol in *Rubicon^{ad-/-}* mice.

Fig. S6c

Comment

5. Metabolic characterization of Rubicon KO mice is inadequate in terms of the changes in TG metabolism shown. Adipose TG breakdown (lipolysis) increases during fasting and under these conditions, significant lipid is mobilized from adipocytes to the liver. Given that this phenomenon is replicated in the adipose Rubicon KO mice under basal state, it is possible that lipophagy/lipolysis (lysosomal and neutral) and fat mobilization are hyperactivated in adipose tissue of Rubicon KO mice. This should be characterized to provide a more comprehensive view of this model. Are lipolytic pathways induced in adipose tissue? Whole body energy expenditure rates (as shown in supplemental data) may not be sensitive enough to pick up depot-specific increases in lipolysis/fat oxidation. It would be particularly useful to carry out these experiments in both fasted Rubicon and Rubicon/Atg5 dual KO mice.

Reply-5

We appreciate the valuable comment. We realised that it is inevitable to determine whether upregulation of lipolysis or lipophagy contributes to the phenotypes of *Rubicon^{ad-/-}* mice. With the additional results, we confirmed that lipolysis or

lipophagy in adipocytes is not a major contributor to the phenotypes. To test the hypothesis that *Rubicon* knockout in adipocytes upregulates lipolysis, we performed qRT-PCR assay, and found that the gene expression of ATGL and HSL tended to be decreased, at least not changed in adipose tissue of *Rubicon^{ad-/-}* mice (Supplementary Fig. 6f). Perilipin1, which promotes lipolysis, in adipocytes was transcriptionally downregulated in *Rubicon^{ad-/-}* mice (Supplementary Fig. 6a, b). Furthermore, we checked plasma NEFA, a product of lipolysis, in both fed and 24-hour-fasted *Rubicon^{ad-/-}* mice. As a result, *Rubicon* knockout in adipocytes did not increase the plasma levels of NEFA under either fed or fasted conditions (Supplementary Fig. 6g). These results indicate that lipolysis is not induced in *Rubicon^{ad-/-}* mice. In addition, *ex vivo* culture of adipose tissue with a lysosomal inhibitor did not increase the protein level of Perilipin1 and Perilipin2, which are potential substrates of lipophagy (Supplementary Fig. 6h), suggesting that lipophagy in WAT does not occur at a detectable level. On the basis of these findings, we conclude that lipolysis and lipophagy are not major contributors to the phenotypes of *Rubicon^{ad-/-}* mice.

Fig. S6a

Fig. S6b

Fig. S6f

Fig. S6g

Fig. S6h

Comment

6. What is the effect of a high fat diet feeding on adipose-specific Rubicon KO mice?

Reply-6

Feeding of an HFD has no effect on the phenotypes of *Rubicon^{ad-/-}* mice. We fed *Rubicon^{ad-/-}* mice an HFD, and found that adipose-specific *Rubicon* knockout mice exhibited a reduction in body weight gain and adipose mass on an HFD (Supplementary Fig. 4a–e). As well as on an NCD, *Rubicon* knockout increased glucose intolerance and hepatic steatosis along with the reduction of plasma adiponectin on an HFD (Supplementary Fig. 4f–i). The phenotypes of obese *Rubicon^{ad-/-}*

$\frac{-}{-}$ mice were cancelled by additional knockout of *Atg5* (Supplementary Fig. 4a–i). Together, these results indicate that Rubicon in adipocytes maintains glucose and lipid metabolism by modulating autophagy independently of diet.

Fig. S4a

Fig. S4b

Fig. S4c

Fig. S4e

Fig. S4g

Fig. S4d

Fig. S4f

Fig. S4h

Fig. S4i

Comment

7. Does loss of Rubicon lead to adipocyte death or defects in adipocyte proliferation? Some of these assessments will greatly help understand this mouse model of hyperactive autophagy.

Reply-7

Thank you for this point. Because previous studies showed that autophagy can promote cell death including apoptosis or necroptosis (J Doherty and EH Baehrecke, *Nat Cell Biol* 2018), we agree with the idea that loss of Rubicon could lead to adipocyte death or defects in adipocyte proliferation via excess autophagy. Therefore, we tested this idea, and found that loss of Rubicon would not result in adipocyte death or adipocyte proliferation defects. While PCNA (a marker for cell proliferation)-positive cells were seen in the epididymis in control mice, PCNA-positive adipocytes were not detected in the eWAT in either control or *Rubicon^{ad-/-}* mice (Supplementary Fig. 7a). This result is consistent with the idea that adipocytes are non-proliferative, and that *Rubicon* knockout does not affect adipocyte proliferation. Furthermore, *Rubicon* knockout in adipocytes did not increase Cleaved Caspase-3 (Supplementary Fig. 7b). TUNEL-positive cells were observed in a DNase I-treated sample, used as a positive control, but not in untreated samples from control or *Rubicon^{ad-/-}* mice (Supplementary Fig. 7c). CLSs, which are dead adipocytes surrounded by macrophages (S Cinti *et al.*, *J. Lipid Res.* 2005), were not detected in WAT of either control or *Rubicon^{ad-/-}* mice, but were seen in that of HFD-fed mice (Supplementary Fig. 7d). These data indicate that adipocyte death is not induced in *Rubicon^{ad-/-}* mice. We conclude that adipocyte death or adipocyte proliferation defects would not contribute to the phenotypes of *Rubicon^{ad-/-}* mice.

Fig. S7a

Fig. S7b

Fig. S7c

Fig. S7d

Comment

8. Concerns related to differentiation experiments: In vitro assessments shown in Fig 3 are also rather preliminary, and it would be more useful to look at changes in autophagy flux and Rubicon levels on day 0, 2, 4, 6, 8, 10, 12 and 16 after initiation of differentiation – revealing details as to when Rubicon expression begins in course of differentiation. On day 10, adipocytes already show significant increase in markers of terminal differentiation, e.g., FAS and ap2, and it is possible that Rubicon is only expressed after the adipocyte is differentiated. This was perhaps the rationale for the authors to choose the adiponectin-Cre line to delete Rubicon in mature adipocytes. As a consequence, data showing impaired differentiation in mature adipocytes knocked out for Rubicon (using the adiponectin-Cre) is rather confusing and difficult to interpret. Furthermore, there are studies that have shown that autophagy is required for muscle and adipocyte differentiation, while this work would suggest that loss of Rubicon, and as a consequence, hyperactivation of autophagy impairs differentiation. How do the authors explain these findings?

Reply-8

We really appreciate this point and apologise for the complicated explanation. We have checked the levels of Rubicon and adipogenesis markers during adipocyte differentiation. But, we found that Rubicon was already expressed on day 0 and increased with adipogenesis while both of FABP4 (aP2) and Adiponectin were fully expressed on day4 to day10 (Fig. 4a, b). This result is not the rationale for us to choose the *Adipoq*-Cre line to delete Rubicon in mature adipocytes. Thus, we removed the claim that *Adipoq*-Cre mediates mature adipocyte-specific knockout. Instead, we simply mentioned that *Adipoq*-Cre mediates adipocyte-specific knockout. The remaining concern is how excess autophagy impairs adipogenesis. As stated below (Fig. 5), activation of PPAR γ restores adipocyte dysfunction by loss of Rubicon. In addition, previous studies showed that SRC-1 and TIF2 in adipocytes work as coactivators of PPAR γ (F Picard *et al.*, *Cell* 2002), and that a protein in the same family, NCoA4, is degraded by autophagy (JD Mancias *et al.*, *Nature* 2014). Thus, we hypothesised that SRC-1 and TIF2 in adipocytes are degraded by excess autophagy, and their reductions lead to a decline in PPAR γ activity and adipocyte function in *Rubicon*^{ad-/-} mice. We tested our hypothesis, and found that SRC-1 and TIF2 in adipocytes were significantly downregulated in *Rubicon*-knockdown cells or *Rubicon*^{ad-/-} mice (Fig. 6a, c), and that SRC-1 and TIF2 in adipose tissue were accumulated with a lysosomal inhibitor (Fig. 6a, b). Importantly, their reductions in *Rubicon*^{ad-/-} mice were cancelled by additional knockout of *Atg5* (Fig. 6d). These results suggest that SRC-1 and TIF2 in adipocytes are degraded by autophagy, and are downregulated in *Rubicon*^{ad-/-} mice. Moreover, we found a potential LIR/GIM in both of SRC-1 and TIF2 (Fig. 7a), and discovered that GABARAP family proteins were immunoprecipitated with SRC-1 or TIF2 in a LIR/GIM-dependent fashion (Fig. 7b, c). Their accumulation with a lysosomal inhibitor or their reduction in *Rubicon*-knockdown cells was dependent on LIR/GIM (Fig. 7d–g). Overexpression of both the LIR/GIM mutant SRC-1 and TIF2 rescued the reduction in adipogenic gene expressions by *Rubicon* knockdown (Fig. 7h). These results collectively indicate that excess autophagy in adipocytes reduces SRC-1 and TIF2, and inhibits PPAR γ activity and adipogenesis in *Rubicon*^{ad-/-} mice. Together with a previous study showing that autophagy deficiency inhibits adipogenesis, we conclude that an optimal level of autophagy is crucial for adipogenesis.

Fig. 4a

Fig. 4b

Fig. 6a

Fig. 6b

Fig. 6c

Fig. 6e

Fig. 6d

Fig. 6f

Fig. 7a

SRC-1	
H. sapiens	286-TGWEDL-291
M. musculus	286-TGWEDL-291
G. gallus	284-TGWEDL-289
X. tropicalis	285-LGWEDL-290
D. rerio	293-PGWEDL-298
TIF2	
H. sapiens	294-PGWEDL-299
M. musculus	294-PGWEDL-299
G. gallus	292-PGWEDL-297
X. tropicalis	288-PGWEDL-293
D. rerio	293-PGWEDL-298

Fig. 7b**Fig. 7c****Fig. 7d****Fig. 7e****Fig. 7f****Fig. 7g****Fig. 7h**
Comment

9. The rationale to look at PPAR agonists in Rubicon KO mice remains unclear.

Reply-9

We are very sorry for the confusing explanation in the previous manuscript. We used a PPAR γ agonist TZD to determine whether activation of PPAR γ recovers the defect in adipocyte function and its associated metabolic disorders, in *Rubicon^{ad-/-}* mice. Because of a strong correlation between the phenotypes of *Rubicon^{ad-/-}* mice and *Pparg*-knockout mice mediated by *Adipoq*-Cre, which show adipose mass reduction, dyslipidemia, hepatic steatosis, glucose intolerance, and endocrine dysfunction (F Wang *et al.*, *PNAS* 2013), we hypothesised that PPAR γ activity is reduced in *Rubicon^{ad-/-}* mice. Indeed, we found that both of *Rubicon*-knockdown cells and *Rubicon^{ad-/-}* mice exhibited a reduction in adipogenic gene expressions (Fig. 4f, g), which are positively regulated by PPAR γ . To test the hypothesis, we treated *Rubicon*-knockdown cells with TZD, and found that TZD treatment recovered the reduction of Adiponectin protein level, adipogenic gene expressions, triglyceride content, and LD area in *Rubicon*-knockdown cells (Fig. 5a–d). TZD treatment also rescued the decline in plasma adiponectin, leptin, and fat mass in *Rubicon^{ad-/-}* mice (Fig. 1d, 5e–g). These results indicate that activation of PPAR γ reverses adipocyte dysfunction by loss of Rubicon. We explained about this discussion in our revised manuscript (Line number: 260–273).

Fig. 4f**Fig. 4g****Fig. 5a****Fig. 5b****Fig. 5c****Fig. 5d****Fig. 5e****Fig. 5f****Fig. 1d****Fig. 5g**
Comment

10. Blots in some of the figure panels should have been run in the same gel.

Reply-10

We have revised this point (Fig. 4a, b, e).

Fig. 4a

Fig. 4b

Fig. 4e

Peer Review File - Reviewers' comments second round:

Reviewers' comments:

Reviewer #1 (Remarks to the Author):

The authors have issued a rebuttal to the previous decision, and in my opinion, have answered many of the issues addressed in the first round of reviews. The revised manuscript delves deeper and offers more information on the role of autophagy inhibition in adipogenesis. There are however further questions and improvements that can be made in this manuscript:

Major concerns:

1. While the use of a single cre to delete 2 floxed genes is theoretically possible, I think the authors need to demonstrate the efficiency in WAT and BAT. S1 demonstrates the deletion in single knockouts, but not in double knockouts. Further S1b shows that the Rubcn ad-/- mouse has a Rubcn WT allele?
2. Throughout the paper, the authors compare Atg5 ad-/- to Rubcn; Atg5 ad-/- mice on a graph separate from the Control v Rubcn ad-/- mice. Is there a statistical difference between control and Atg5 ad-/- mice? Are the control mice littermates?
3. Ponceau images are difficult to see and should be replaced with a housekeeping protein or other loading control.
4. How do adipocytes react to autophagic stimuli, such as rapamycin or starvation?
5. The authors should also examine the LC3-II to LC3-I ratio.
6. The authors make claims about increases in protein levels (4b, 4d, 7d, 7e, 7g) that do not seem to be dramatic or even visible by Western. Would these changes be more evident in a "stressed" system, like rapamycin treatment?
7. While the authors have done some studies to rule out mitophagy, I do not think they have fully explored the vital role of mitochondria in this scenario. The authors should look at Parkin, PINK1, and use mito-specific dyes, such as mitotracker, etc. to examine the health of the mitochondria in these models. It may be helpful to examine mitochondrial respiration as well.
8. Flow cytometry should be performed to assess the infiltration of immune cells into tissues, rather than qPCR.

Minor comments:

1. The histological images are difficult to see.
2. The authors often fail to mention key components of figures in the text or mention figures out of order.
3. Statistics for 1g and 3h-i? Is Figure 2c (right) significant or not?

Reviewer #3 (Remarks to the Author):

The finding that autophagic degradation of PPARgamma coactivators SRC-1 and TIF2 was resulted from reduced Rubicon suggests a new pathway for fat tissue atrophy and metabolic disorders observed with aging. Although this study has potential interests, the mechanistic understanding for the regulation of rubicon expression will strengthen the biological significance of autophagy in adipose tissue upon aging. In the revised manuscript, the authors have performed several experiments to address the comments from the Reviewer #2. Nonetheless, following issues were not properly answered by the authors.

1. The age of mice used for qPCR data in Reply-1 was younger than that of mice used in p62 western blot of main Figure1. The decreasing tendency of p62 transcripts in 18-month-old mice is

likely to decrease significantly when observed in 25-month-old mice. This should be appropriately addressed by experiments.

2. In Reply #2, the comparison of the autophagy flux was done in eWAT and iBAT. The reviewer #2 asked to compare the autophagy flux in eWAT and iWAT. This should be performed.

3. In addition to showing lipolytic gene expression, functional aspects such as ex vivo glycerol release assay will support Reply #5.

4. To clarify the degree of lipophagy and mitophagy, p62 pull down-LC3 II measurement can be further conducted.

Point-by-point response

Reviewer #1 (Remarks to the Author):

General comments

The authors have issued a rebuttal to the previous decision, and in my opinion, have answered many of the issues addressed in the first round of reviews. The revised manuscript delves deeper and offers more information on the role of autophagy inhibition in adipogenesis. There are however further questions and improvements that can be made in this manuscript:

We appreciate the insightful and constructive comments. Your suggestions have greatly helped to improve our study. Following is the detailed response to your comments. We have highlighted the changes in the revised manuscript using blue color.

Major

Comment-1

While the use of a single cre to delete 2 floxed genes is theoretically possible, I think the authors need to demonstrate the efficiency in WAT and BAT. S1 demonstrates the deletion in single knockouts, but not in double knockouts. Further S1b shows that the Rubcn ad^{-/-} mouse has a Rubcn WT allele?

Reply-1

Thank you for the comment. We conducted the immunoblot to test the knockout efficiency in both the eWAT and iBAT from the double-knockout mice, and confirmed that the double knockout mice show protein levels of both Rubicon and Atg5, each of which was comparable to that of single knockout mice of *Rubicon* or *Atg5* (Supplementary Fig. 1c, d). We are sorry for the error in Figure S1b. We have revised this point to avoid any confusion (Supplementary Fig. 1b).

Fig. S1c

Fig. S1d

Fig. S1b

Comment-2

Throughout the paper, the authors compare *Atg5* ad-/- to Rubcn; *Atg5* ad-/- mice on a graph separate from the Control v Rubcn ad-/- mice. Is there a statistical difference between control and *Atg5* ad-/- mice? Are the control mice littermates?

Reply-2

Thank you for the suggestion. This is very informative. It is also our interest whether *Atg5* deficiency causes any changes in adipocytes. Because the major focus of this report is to determine if upregulation of autophagy is a cause of adipocyte dysfunction, we primarily focused on comparing control mice vs *Rubicon*^{ad-/-} mice, or *Atg5*^{ad-/-} mice vs *Rubicon*^{ad-/-}; *Atg5*^{ad-/-} mice. Namely, we utilized an *Atg5*-deficient background to examine the autophagy-dependency of the phenotype in *Rubicon*^{ad-/-} mice. Also, all mice used in this study were not littermates because we used *Adipoq-Cre* mice as controls to exclude the possibility that the phenotypes of the knockout mice arise from *Cre*-recombinase-mediated cellular dysfunction (A Loonstra *et al.*, *PNAS* 2001). As the number of mice used for the study should be minimized according to the guideline for animal experiment from the Institutional Committee of Osaka University, we generated *Rubicon*^{ad-/-} mice by crossing *Rubicon*^{fllox/fllox}; *Adipoq-Cre* mice with *Rubicon*^{fllox/fllox} mice. All kinds of transgenic mice were initially backcrossed into the C57BL/6J wild-type strain at least six times, and each mouse was generated from the same mouse colony maintained in the C57BL/6J background. We added this information to the revised manuscript (Line number: 455–460).

Considering the readers' interest in the phenotype of *Atg5* deficiency in adipocytes, we have updated the statistical information on the several data shown here. Indeed, we found that *Atg5*^{ad-/-} mice showed a reduction in plasma cholesterol (Fig. 2b) and leptin (Fig. 2h), and an increase in hepatic cholesterol (Fig. 2e). These results suggest that an autophagy-dependent or -independent role of ATG5 in adipocytes could be important for Leptin secretion or cholesterol metabolism. Interestingly, HFD-fed *Atg5*^{ad-/-} mice exhibited a reduction in visceral eWAT and an increase in subcutaneous iWAT and iBAT, compared with the control counterparts (Supplementary Fig. 4b). This fat maldistribution was associated with a reduction in plasma adiponectin (Supplementary Fig. 4i), and an increase in obesity-induced glucose intolerance (Supplementary Fig. 4f) and hepatic triglyceride accumulation (Supplementary Fig. 4h). The *Atg5* deficiency in adipocytes also promoted formation of the HFD-induced crown-like structure (CLS) in the eWAT (Supplementary Fig. 4c, 8d), consistent with a previous report showing the role of ATG3 and ATG16L1 in adipose tissue inflammation during obesity (J Cai *et al.*, *Cell Reports* 2018). These results suggest that ATG conjugation system, which includes ATG3, ATG5, and ATG16L1, in adipocytes could prevent obesity-related metabolic disorders.

Fig. 2b**Fig. 2h****Fig. 2e****Fig. S4b****Fig. S4i****Fig. S4f****Fig. S4h****Fig. S4c****Fig. S8d**
Comment-3

Ponceau images are difficult to see and should be replaced with a housekeeping protein or other loading control.

Reply-3

We agree with this comment. We have replaced the Ponceau-S images with the immunoblots of β -actin or α -tubulin as the loading control. We updated the quantification for each protein according to the improved controls.

Comment-4

How do adipocytes react to autophagic stimuli, such as rapamycin or starvation?

Reply-4

We consider this comment is very informative. Adipocyte function as an energy storage may physiologically decrease in starved condition that simultaneously induces autophagy. Because our data suggest that SRC-1 and TIF2 in adipocytes are degraded by autophagy, starvation may change this regulatory system as well. We found that SRC-1 and TIF2 were significantly decreased in the adipose tissue in mice fasted for 48 hours (Fig. 6g). It was also the case of in vitro experiments showing that the starved 3T3-L1 adipocytes show a time-dependent reduction in both proteins (Fig. 6h). It was consistent with the fact that most of adipogenic gene expressions in 3T3-L1 cells were reduced with starvation (Fig. 6i), suggesting the starvation-induced decline of adipocyte function. Thus, we concluded that the declines in SRC-1 and TIF2 in adipocytes are shared features of the decline in adipocyte function in both starvation and aging situation that are associated with the increase in autophagy.

Fig 6g

Fig 6h

Fig 6i

Comment-5

The authors should also examine the LC3-II to LC3-I ratio.

Reply-5

We have confirmed that the ratio of LC3-II to LC3-I in WAT (Supplementary Fig. 2c) and BAT (Supplementary Fig. 2d) was not significantly changed between control mice and *Rubicon*^{ad-/-} mice. It is not surprising since Rubicon is a negative regulator of autophagy at the late step of autophagy; it regulates the fusion between autophagosome and lysosome. We observed the protein levels of autophagic substrate LC3-II and p62 were decreased in WAT (Supplementary Fig. 2c) and BAT (Supplementary Fig. 2d) of *Rubicon*^{ad-/-} mice. Moreover, our *ex vivo* autophagic flux assay support the increased autophagic activity in WAT (Supplementary Fig. 2e, f) and BAT (Supplementary Fig. 2g) of *Rubicon*^{ad-/-} mice.

Fig. S2c

Fig. S2d

Fig. S2e

Fig. S2f

Fig. S2g

Comment-6

The authors make claims about increases in protein levels (4b, 4d, 7d, 7e, 7g) that do not seem to be dramatic or even visible by Western. Would these changes be more evident in a "stressed" system, like rapamycin treatment?

Reply-6

We agree with this comment that the representative immunoblots did not show drastic changes. We are sorry about this controversy. We consider that this is a technical issue of our experiments and visualizations. To improve the sensitivity of the experiments, we repeated the experiments with increased protein loading for each sample, and observed clearer data (Fig 4b, d, 7f, g). We also updated the quantification as well (Fig 7f, g). In previous Figure 7d and 7e, we used undifferentiated 3T3L1 cells. Therefore, we repeated the experiments using mature 3T3-L1 adipocytes at Day 10, and found that SRC-1 and TIF2 were clearly accumulated with lysosomal inhibitor treatment in mature adipocytes (Fig. 7d, e). These data support our conclusion that SRC-1 and TIF2 are degraded by autophagy in mature adipocytes.

Fig. 4b

Fig 4d

Fig. 7f

Fig. 7g

Fig. 7d

Fig. 7e

Comment-7

While the authors have done some studies to rule out mitophagy, I do not think they have fully explored the vital role of mitochondria in this scenario. The authors should look at Parkin, PINK1, and use mito-specific dyes, such as mitotracker, etc. to examine the health of the mitochondria in these models. It may be helpful to examine mitochondrial respiration as well.

Reply-7

We appreciate the suggestion. First, we assessed the mitochondrial respiration in *Rubicon* knockdown cells using tetramethylrhodamine methyl ester (TMRM), which visualizes the mitochondrial membrane potential. We found that *Rubicon* knockdown did not affect the mitochondrial membrane potential whereas an ionophore valinomycin remarkably decreased the membrane potential as a positive control (Supplementary Fig. 7a). To estimate Parkin-dependent mitophagy in the absence of Rubicon, we treated the cells with valinomycin to see the recruitment of Parkin onto the mitochondria. The recruitment rate of Parkin was not changed in *Rubicon* knockdown cells (Supplementary Fig. 7b). Strikingly, the degradation rate of outer and inner mitochondrial proteins was also not affected by *Rubicon* knockdown (Supplementary Fig. 7d). In addition, the protein levels of Parkin and PINK1 were not significantly changed in the *Rubicon* knockdown cells (Supplementary Fig. 7d), the WAT (Supplementary Fig. 2a) and the BAT (Supplementary Fig. 2b) of *Rubicon*^{ad-/-} mice. These data collectively indicate that downregulation of Rubicon in adipocyte does not affect mitophagy.

Fig. S7a

Fig. S7d

Fig. S7b

Fig. S2a

Fig. S2b

Comment-8

Flow cytometry should be performed to assess the infiltration of immune cells into tissues, rather than qPCR.

Reply-8

We conducted the flow cytometry assay to assess the immune cell infiltration in the adipose tissue, and found that the rate of CD45⁺ hematopoietic cells was not increased in the stromal vascular fraction (SVF) from the eWAT of *Rubicon*^{ad-/-} mice, compared with control mice (Supplementary Fig. 9g). Notably, the rate of CD11b⁺ F4/80⁺ macrophage in the CD45⁺ hematopoietic cells was also not increased (Supplementary Fig. 9h). These data indicate that *Rubicon* knockout in adipocytes does not induce immune cell inflammation.

Fig. S9g

Fig S9h

Minor

Comment-1

The histological images are difficult to see.

Reply-1

Thank you for the suggestion. We replaced them with the enlarged images as more suitable representative images. And we added the arrows that indicate positive or negative staining for PCNA (Supplementary Fig. 8a) or TUNEL (Supplementary Fig. 8c).

Fig. S8a

Fig. S8c

Comment-2

The authors often fail to mention key components of figures in the text or mention figures out of order.

Reply-2

We apologize the confusing explanation. We added the detailed information for the figures mentioned in the text (Line number: 95, 113–117, 124, 125, 136, 142, 143, 147, 150, 152, 167, 168, 199, 200, 206, 233, 289, 290, 295, 296, 301, 316, 356).

Comment-3

Statistics for 1g and 3h-i? Is Figure 2c (right) significant or not?

Reply-3

We are sorry about the lack of statistical information for these data. We added the information for adipocyte area in the NFD-fed mice (Fig. 1g), HFD-fed mice (Supplementary Fig. 4d), 12-month-old mice (Fig. 3h), and 18-month-old mice (Fig. 3i). *Rubicon* knockout significantly decreased the adipocyte size in these mice. We are sorry for the error in previous Figure 2c. There is no difference between *Atg5^{ad-/-}* mice and *Rubicon^{ad-/-}; Atg5^{ad-/-}* mice in two-way ANOVA (Fig. 2c).

Fig. 1g

Fig. S4d

Fig. 3h

Fig. 3i

Fig 2c

Reviewer #3 (Remarks to the Author):

General comments

The finding that autophagic degradation of PPARgamma coactivators SRC-1 and TIF2 was resulted from reduced Rubicon suggests a new pathway for fat tissue atrophy and metabolic disorders observed with aging. Although this study has potential interests, the mechanistic understanding for the regulation of rubicon expression will strengthen the biological significance of autophagy in adipose tissue upon aging. In the revised manuscript, the authors have performed several experiments to address the comments from the Reviewer #2. Nonetheless, following issues were not properly answered by the authors.

Thank you so much for the valuable comments. According to the suggestions, we have carefully planned and conducted several experiments to clarify the concerns and improve the manuscript. Following is the detailed response to the comments. We have highlighted the changes in the revised manuscript using blue color.

Comment

1. The age of mice used for qPCR data in Reply-1 was younger than that of mice used in p62 western blot of main Figure1. The decreasing tendency of p62 transcripts in 18-month-old mice is likely to decrease significantly when observed in 25-month-old mice. This should be appropriately addressed by experiments.

Reply-1

We are sorry for the confusing data. According to the comment, we examined the p62 transcripts in the WAT from 25-month-old mice. As expected, p62 transcripts were significantly decreased in 25-month-old mice (Supplementary Fig. 5a). Hence it could not be clearly suggested that the decrease in p62 protein in adipocytes at 25-month-old mice is due to an upregulated autophagic activity only from this data. However, as shown also in the previous manuscript, a significant reduction in p62 protein levels were already evident even at the 12- and 18-month-old mice (Fig. 3a) whose p62 transcripts were not significantly decreased yet (Fig. 3b). And *ex vivo* autophagic flux assay showed that the autophagic activity in adipose tissue was increased at 18-month-old (Fig. 3c). Taken together, we conclude that autophagy in adipocytes is upregulated with aging. We added explanation about this interpretation to avoid any confusion (Line number: 187–194).

Fig 3a

Fig 3b

Fig. S5a

Fig. 3c

Comment

2. In Reply #2, the comparison of the autophagy flux was done in eWAT and iBAT. The reviewer #2 asked to compare the autophagy flux in eWAT and iWAT. This should be performed.

Reply-2

We apologize for the misunderstanding. To address this issue, we conducted the *ex vivo* autophagy flux assay in iWAT, and found that autophagic activity tended to be increased in iWAT of *Rubicon^{ad-/-}* mice compared with the control (Supplementary Fig. 2f). Together with the data of eWAT (Supplementary Fig. 2e) and iBAT (Supplementary Fig. 2g), the iWAT data strengthens our conclusion that autophagic activity is upregulated in adipose tissues in *Rubicon^{ad-/-}* mice.

Fig. S2e

Fig. S2f

Fig. S2g

Comment

3. In addition to showing lipolytic gene expression, functional aspects such as *ex vivo* glycerol release assay will support Reply #5.

Reply-3

We appreciate the insightful suggestion. *Ex vivo* lipolysis assay showed that catecholamine-induced release of glycerol was not significantly increased in eWAT (Supplementary Fig. 6h), iWAT (Supplementary Fig. 6i), or iBAT (Supplementary Fig. 6j) from *Rubicon*^{ad-/-} mice. That of NEFA was not also increased in *Rubicon*^{ad-/-} mice (Supplementary Fig. 6k-m). Together with our data showing expression of genes involved in lipolysis, we concluded that the loss of *Rubicon* does not enhance lipolysis in adipocytes.

Comment

4. To clarify the degree of lipophagy and mitophagy, p62 pull down-LC3 II measurement can be further conducted.

Reply-4

Thank you for the comment. Because p62 is a potential receptor for lipophagy or mitophagy also in adipocytes, we conducted the immunoprecipitation assay of p62 to see the interaction between p62 and LC3-II in *Rubicon*-knockdown 3T3-L1 adipocytes. As a result, *Rubicon* knockdown did not change the interaction (Supplementary Fig. 7c). In addition, *Rubicon* knockdown did not affect the mitochondrial membrane potential (Supplementary Fig. 7a) and the Parkin-recruitment onto mitochondria (Supplementary Fig. 7b) during mitophagy. Strikingly, the degradation rate of mitochondrial proteins was not significantly changed by *Rubicon* knockdown (Supplementary Fig. 7d). Further, *ex vivo* culture of adipose tissue with a lysosomal inhibitor did not increase the protein levels of LD-resident Perilipin1, Perilipin2, and mitochondrial Complex III subunit Core 1, but did increase LC3-II (Supplementary Fig. 7e). Taken together, we concluded that the downregulation of *Rubicon* does not cause any significant changes in selective autophagy in adipocytes.

Fig. S7c

Fig. S7a

Fig. S7b

Fig. S7d

Fig. S7e

Peer Review File - Reviewers' comments third round:

Reviewer #1 (Remarks to the Author):

I think overall the authors have done a good job addressing the concerns raised in the rebuttal. I do think more clarification should be given about the lack of littermates, specifically:

1. Were the controls used (Cre+) maintained in the same mouse colony and same space?
2. Were animals from the Cre+ control colony used to generate the KO and DKO strains?
3. Were animals co-housed for the aging and HFD studies? If so, were they co-housed by genotype?

Also, the food formulation for the HFD should be included.

Reviewer #3 (Remarks to the Author):

In this revised manuscript, the authors provided both new experiments and substantial textual revisions, which address most concerns. They showed that the reduction of Rubicon in adipose tissue of aged mice is commonly observed among fat depots. In addition, they provided several evidence that the downregulation of Rubicon did not cause significant changes in selective autophagy (lipophagy and mitophagy) in adipocytes.

Point-by-point response

Reviewer #1 (Remarks to the Author):

General comments

I think overall the authors have done a good job addressing the concerns raised in the rebuttal. I do think more clarification should be given about the lack of littermates, specifically:

We really appreciate your insightful and constructive suggestions since the first review. Your comments have greatly improved our manuscript.

Comment-1

Were the controls used (Cre+) maintained in the same mouse colony and same space?

Reply-1

Yes. The control mice were maintained in the same mouse colony and same space. We added this information to the revised manuscript (Line number: 490–491).

Comment-2

Were animals from the Cre+ control colony used to generate the KO and DKO strains?

Reply-2

Yes. The control colony was used for the generation of the knockout strains. We added this information to the revised manuscript (Line number: 491).

Comment-3

Were animals co-housed for the aging and HFD studies? If so, were they co-housed by genotype?

Reply-3

Yes. Just after weaning, age-matched mice were randomly co-housed independently from genotype, for the aging and HFD studies. No fighting was observed in the mice. We added this information to the revised manuscript (Line number: 491–493).

Reviewer #3 (Remarks to the Author):

General comments

In this revised manuscript, the authors provided both new experiments and substantial textual revisions, which address most concerns. They showed that the reduction of Rubicon in adipose tissue of aged mice is commonly observed among fat depots. In addition, they provided several evidence that the downregulation of Rubicon did not cause significant changes in selective autophagy (lipophagy and mitophagy) in adipocytes.

Thank you so much for giving us the valuable comments. Your suggestions have helped us to make this study better.